# Mpf1 affects the dual distribution of tail-anchored proteins between mitochondria and peroxisomes

Nitya Aravindan[1], Daniela G Vitali [1], Julia Breuer [1], Jessica Oberst[1], Einat Zalckvar [2,3], Maya Schuldiner [2] & Doron Rapaport [1✉]

## Abstract

**Most cellular proteins require targeting to a distinct cellular compartment to function properly. A subset of proteins is distributed to two or more destinations in the cell and little is known about the mechanisms controlling the process of dual/multiple targeting. Here, we provide insight into the mechanism of dual targeting of proteins between mitochondria and peroxisomes. We perform a high throughput microscopy screen in which we visualize the location of the model tail-anchored proteins Fis1 and Gem1 in the background of mutants in virtually all yeast genes. This screen identifies three proteins, whose absence results in a higher portion of the tail-anchored proteins in peroxisomes: the two paralogues Tom70, Tom71, and the uncharacterized gene YNL144C that we rename mitochondria and peroxisomes factor 1 (Mpf1). We characterize Mpf1 to be an unstable protein that associates with the cytosolic face of the mitochondrial outer membrane. Furthermore, our study uncovers a unique contribution of Tom71 to the regulation of dual targeting. Collectively, our study reveals, for the first time, factors that influence the dual targeting of proteins between mitochondria and peroxisomes.**

**Keywords** Dual Targeting; Fis1; Mitochondria; Peroxisomes; Tail-anchored Proteins
**Subject Categories** Membranes & Trafficking; Organelles

## Introduction

Eukaryotic cells have evolved complex machineries that direct, with the help of targeting signals, cytosolically synthesized proteins to specific intracellular locations. While most proteins target to a single compartment, there are those that are targeted to two (or even more) cellular destinations. Some examples of such proteins include the metabolic enzymes fumarase and aconitase, where in addition to the mitochondrial population a certain portion of the protein is also found in the cytosol or even the nucleus (Regev-Rudzki and Pines, 2007; Stein et al, 1994; Yogev et al, 2011).

Several cases of such dually targeted membrane proteins are those that distribute between mitochondria and peroxisomes. These two organelles maintain extensive crosstalk and are transiently associated by multiple contact sites. In yeast, Pex11, a key protein involved in initiating peroxisome division was found to interact with Mdm34, a constituent of the ER-mitochondria encounter structure (ERMES) complex, thus mediating these mitochondria-peroxisome contact sites (Mattiazzi Ušaj et al, 2015). Furthermore, peroxisomes were found in close proximity to sites enriched in pyruvate dehydrogenase (PDH) complex responsible for acetyl-CoA synthesis in the mitochondrial matrix, further validating the organelles' co-dependency in their metabolic processes (Cohen and Schuldiner, 2011; Mattiazzi Ušaj et al, 2015; Shai et al, 2016). In a high throughput screen, Fzo1 and Pex34 were found to contribute to the formation of mitochondria-peroxisomes tethers (Shai et al, 2016). In addition to their close physical and metabolic relationships, mitochondria and peroxisomes also utilize many similar outer membrane proteins, such as the same fission machinery, for their division.

A central component of this fission machinery is Fis1, a tail-anchored (TA) protein that can be targeted to both mitochondrial and peroxisomal membranes in yeast, plants, and mammalian cells (Koch et al, 2005; Kuravi et al, 2006; Schrader et al, 2016). The adapter protein Mdv1, which is recruited by Fis1 to the site of fission, engages in turn the dynamin-like protein Dnm1 that mediates the final fission step (Bleazard et al, 1999; Mozdy et al, 2000; Shaw and Nunnari, 2002; Yoon et al, 2003). Notably, mammalian homologs of the fission components Fis1 (hFIS1 in mammals) and Dnm1 (DRP1 in mammals) as well as Mitochondrial Fission Factor (MFF), which is found only in metazoans, were also found to be dually localized to mitochondria and peroxisomes (Schrader et al, 2016). Combined defects in the organellar fission have been linked to several pathophysiological conditions and therefore it is important to understand the biogenesis of proteins involved in these cellular machineries (Ong et al, 2013; Schrader et al, 2022). Additional examples for proteins dually localized to both mitochondria and peroxisomes are the TA protein Gem1 (MIRO1 in mammals) and the ATPase Msp1 (ATAD1 in mammals) (Costello et al, 2017; Okreglak and Walter, 2014).

While a clear mechanism for targeting TA proteins to the secretory pathway has been worked out (Schuldiner et al, 2008; Stefanovic and Hegde, 2007), the mechanism involved in the

[1]Interfaculty Institute of Biochemistry, University of Tübingen, Tübingen, Germany. [2]Department of Molecular Genetics, Weizmann Institute of Science, Rehovot, Israel. [3]The Mina and Everard Goodman Faculty of Life Sciences, Bar-Ilan University, Ramat-Gan, Israel. ✉E-mail: doron.rapaport@uni-tuebingen.de

correct targeting of TA proteins to mitochondria is still widely unknown. It has been previously shown that optimal hydrophobicity of the transmembrane domain and the presence of charged residues is essential for the correct targeting of TA proteins to mitochondria and/or peroxisomes (Bittner et al, 2022; Borgese et al, 2007; Costello et al, 2017). While the mitochondrial import (MIM) complex was suggested to promote the biogenesis of the TA proteins Fis1 and Gem1 (Doan et al, 2020), other studies have also reported insertion of Fis1 to the mitochondrial outer membrane (MOM) and to lipid vesicles in an unassisted manner (Krumpe et al, 2012, Vitali et al, 2020).

For peroxisomal TA proteins, a dedicated pathway for membrane targeting—mediated by Pex19 and Pex3 exists (Fujiki et al, 2006). Nascent TA proteins with a peroxisomal membrane targeting signal (mPTS) are recognized by Pex19 in the cytoplasm and delivered to the peroxisomal membrane by interacting with the membrane receptor Pex3 (Chen et al, 2014; Götte et al, 1998). Surprisingly, depletion of *PEX19* resulted in reduced steady state levels of Fis1 and Gem1 also in mitochondrial fractions, suggesting an unexpected contribution of Pex19 to the biogenesis of mitochondrial TA proteins (Cichocki et al, 2018). Hence, despite some understanding of how TA proteins arrive to either mitochondria or peroxisomes, how the distribution of such dually localized proteins is regulated is still quite puzzling.

To obtain new insights on the mechanisms that control dual targeting of proteins to both mitochondria and peroxisomes and to unravel novel factors involved in this process, we used a high-throughput visual screen with fluorescently labeled TA proteins in the bakers' yeast *Saccharomyces cerevisiae*. We found that the deletion of the uncharacterized gene *YNL144C* (re-named in this study as mitochondria and peroxisomes factor 1 (*MPF1*)), as well as each of the paralogous genes, *TOM70* and *TOM71* led to an enhanced localization of Fis1 to peroxisomes. Accordingly, over-expressing Tom71 caused Fis1 to localize more to mitochondria. We further characterized Mpf1 and identified it as an unstable protein on the surface of mitochondria controlled by the presence of Tom70 and Tom71. Collectively, our findings describe the involvement of Mpf1, Tom70, and Tom71 in regulating the dual distribution of Fis1 and Gem1 to mitochondria and peroxisomes.

## Results

### A high throughput visual screen reveals candidates that affect the dual distribution of TA proteins to mitochondria and peroxisomes

Some yeast TA proteins like Fis1 and Gem1 are found on both organelles—mitochondria and peroxisomes. To identify factors that influence the dual distribution of these proteins, we decided to employ a high-throughput microscopy screen. To visualize the TA proteins of interest, we generated strains expressing mCherry tagged versions of either Fis1 or Gem1 on the background of a peroxisomal marker, Pex3-GFP. Then, using an automated procedure (Cohen and Schuldiner, 2011), we introduced these two tagged proteins into the background of a collection containing deletion strains of all the yeast non-essential genes and depletion strains of all the essential ones (Fig. 1A). Thus, a new collection of deletion/depletion strains expressing the mCherry tagged TA

protein (Fis1 or Gem1) along with GFP tagged Pex3 was created. This new collection was then subjected to a high-throughput microscopy screen to identify those strains where the distribution of the TA proteins Fis1 and Gem1 between mitochondria and peroxisomes is altered (Fig. 1A).

While many proteins altered the distribution of the TA proteins between the two organelles, it appeared that this was often secondary to biogenesis and/or morphological defects of the respective organelles. Hence, we decided to consider as a real hit only strains that fulfilled two criteria: (i) have normal biogenesis of peroxisomes (as reflected by the number of GFP puncta structures per cell), and (ii) display normal mitochondrial morphology, as observed with the mCherry-tagged TA protein. Considering these requirements, the screen allowed us to identify several proteins that might influence the distribution of Fis1 and Gem1 to peroxisomes (see Dataset EV1 for the full list). Further manual examination of the complete list led us to focus on the uncharacterized protein Ynl144c (that we re-named as mitochondrial and peroxisomal factor 1, Mpf1), and the two paralogue proteins Tom70 and Tom71. The absence of each of these three proteins led to a greater co-localization of mCherry-Fis1 with Pex3-GFP stained peroxisomes compared to control cells (Fig. 1B,C). When we quantified this phenotype, we could detect co-localization of mCherry-Fis1 with 20% of peroxisomes in the wild type (WT) cells and this number was considerably increased in cells lacking Mpf1, Tom70, or Tom71 (Fig. 1C).

To investigate whether this observation was limited only to mCherry-Fis1, we quantified the co-localization of mCherry-Gem1 with Pex3-GFP and observed the same trend, indicating that the identified proteins have a general effect on dually distributed TA proteins (Fig. EV1). We further wished to confirm that these observations were not the result of fragmented mitochondria that were misinterpreted for peroxisomal puncta. To this aim, we visualized mitochondrial morphology by staining the organelle with Om45-GFP and observed that the mitochondrial morphology was not altered in *mpf1Δ* and *tom71Δ* cells as compared to control cells (*tim13Δ*) (Fig. EV2). We noticed a slightly altered mitochondrial morphology in *tom70Δ* cells (Fig. EV2), which is not surprising considering the functions of Tom70 as an important mitochondrial import receptor and a docking site for cytosolic chaperones (Backes et al, 2021; Kreimendahl and Rassow, 2020; Yamamoto et al, 2009; Young et al, 2003). Collectively, the visual screen identified Mpf1, Tom70, and Tom71 as potential factors affecting the dual distribution of TA proteins.

### Physical separation of mitochondria and peroxisomes validates the involvement of the identified hits in the dual distribution of Fis1

To confirm by another unrelated approach that the candidates that we picked by the visual screen truly affect the dual distribution of Fis1 to mitochondria and peroxisomes, we monitored the distribution of Fis1 by subcellular fractionation. To obtain optimal separation, the cells were grown on oleate as a carbon source, a condition known to induce proliferation of peroxisomes. After obtaining a crude mitochondrial fraction, we used centrifugation of a Histodenz gradient to separate mitochondria from peroxisomes and 12 fractions of the gradient were collected. Our protocol could nicely differentiate between the two organelles despite their strong

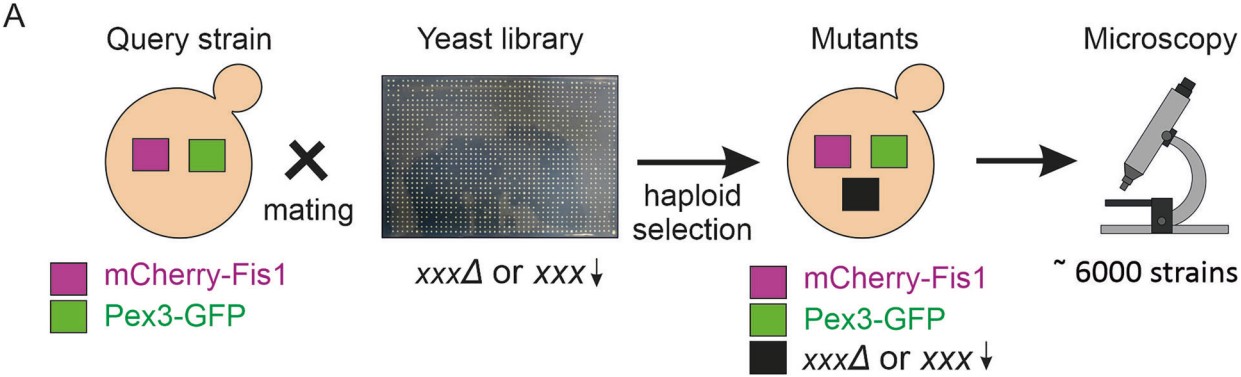

A

Query strain / Yeast library / Mutants / Microscopy

mCherry-Fis1
Pex3-GFP

× mating

xxxΔ or xxx↓

haploid selection

mCherry-Fis1
Pex3-GFP
xxxΔ or xxx↓

~ 6000 strains

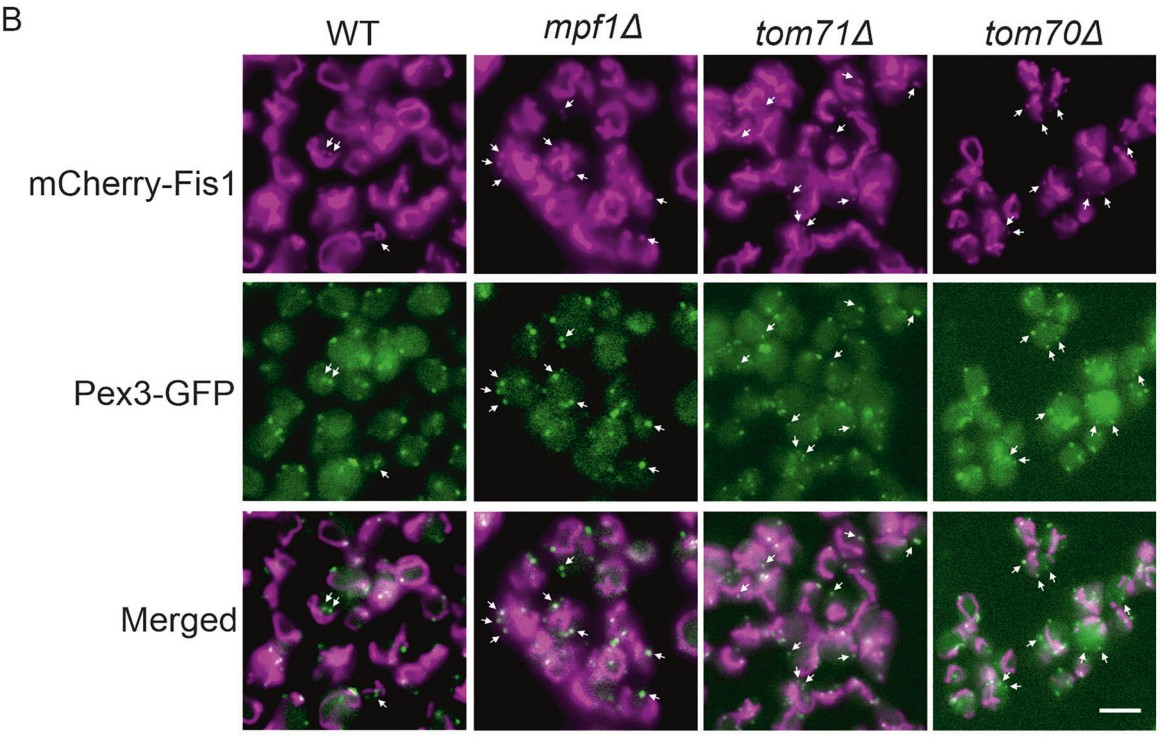

B

mCherry-Fis1

Pex3-GFP

Merged

WT          mpf1Δ          tom71Δ          tom70Δ

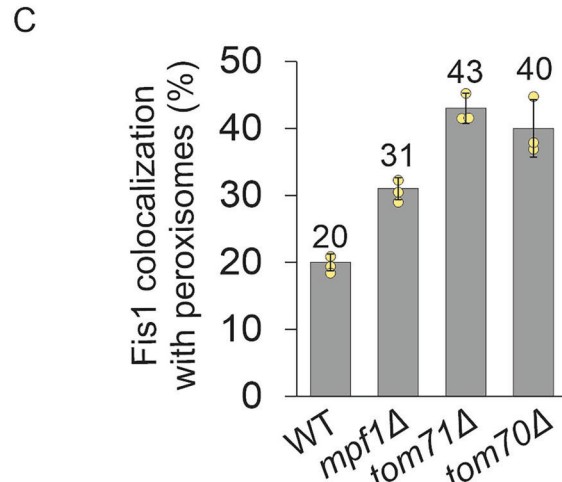

C

Figure 1.   A high-throughput microscopy screen reveals proteins that affect dual distribution of Fis1.

(**A**) Illustration of the screen aiming to find factors that affect dual targeting of Fis1. mCherry-Fis1 and Pex3-GFP were integrated into yeast deletion and depletion libraries. The resultant strains, each containing a unique gene deletion/depletion and carrying the fluorescently labeled target proteins, were visualized using automated microscopy. (**B**) Representative images of WT and three deletions strains with altered distribution of Fis1. The phenotype was observed by detecting co-localization of mCherry-Fis1 with Pex3-GFP (shown with white arrows). Scale bar, 5 μm. (**C**) Quantification of the co-localization of mCherry-Fis1 with peroxisomes. Total number of peroxisomes (visualized by Pex3-GFP) were counted in 100 cells in each of three independent experiments. Subsequently, the percentage of mCherry-Fis1 puncta co-localized with peroxisomes was determined. The graph represents the average of three independent experiments, error bars represent ±standard deviation (S.D.).

physical associations: Tom20, a bona fide mitochondrial marker protein was enriched in the first four fractions (lanes 1–4), whereas the peroxisomal marker protein Pex14 was found preferentially in the last four fractions (9–12). As expected for a dually targeted protein, Fis1 was present in both sets of fractions (lanes 1–4 and 9–12) (Fig. 2A). Fis1 levels in the various fractions were quantified and the total amount in fractions 1–4 was considered as mitochondrial Fis1 whereas the material in fractions 9–12 was counted as peroxisomal population.

Using this approach, we compared the distribution of native Fis1 in wild type (WT) cells to that in the mutated strains. Whereas in WT cells, 70% of Fis1 were found to be mitochondrial, this portion decreased to 57% and 52% in *mpf1Δ* and *tom71Δ* cells, respectively (Fig. 2A,B). In parallel to the decrease in mitochondrial Fis1, we observed an increase in the peroxisomal portion of the protein. We found 23% of Fis1 in peroxisomes in WT cells whereas this fraction had increased to 34% and 37% in *mpf1Δ* and *tom71Δ* cells, respectively (Fig. 2A,B). Unfortunately, we could not analyze the distribution of Fis1 in *tom70Δ* cells because, for unknown reasons, the separation of mitochondria from peroxisomes did not work well with cells from this strain. Hence, this technique could not validate Tom70 as a factor regulating the dual distribution of TA proteins.

We were then interested to explore the distribution phenotype upon the parallel deletion of both *MPF1* and *TOM71*. Surprisingly, we observed that in *tom71Δ/mpf1Δ* double deletion cells, the alteration in the Fis1 distribution was less profound than in the corresponding single deletion strains (Fig. 2A,B). It might be that loss of both Mpf1 and Tom71 leads to compensatory upregulation of alternative factors to restore mitochondrial Fis1 levels. Taken together, the physical separation of mitochondria from peroxisomes confirmed the (direct or indirect) involvement of Mpf1 and Tom71 in regulating the dual distribution of Fis1 between mitochondria and peroxisomes.

## Tom71 has a unique role in Fis1 distribution, setting it apart from Tom70

Tom71 is a paralogue of Tom70, sharing 53% sequence identity, whose abundance is rather low—only about 10% of the levels of Tom70 (Morgenstern et al, 2021; Schlossmann et al, 1996). Tom70 plays a pivotal role as a mitochondrial import receptor and chaperones' docking site and is required for the biogenesis of many mitochondrial proteins (Backes et al, 2021; Kreimendahl and Rassow, 2020; Yamamoto et al, 2009; Young et al, 2003). Tom70 and Tom71 are thought to have overlapping functions, and a specialized role unique to Tom71 has not been reported yet.

Our work suggest a unique role for Tom71 since its deletion led to a distribution phenotype even in the presence of Tom70. To

better understand the involvement of both proteins in the dual distribution of Fis1, we created strains where each of the protein is overexpressed. To that aim, the endogenous promoter of *TOM70* in WT, *mpf1Δ* and *tom71Δ* cells was replaced by the strong *GPD* promoter, resulting in a dramatic overexpression (Fig. 3A). We then separated mitochondria and peroxisomes and quantified mitochondrial and peroxisomal levels of Fis1 (Fig. 3B,C). We observed that elevated levels of Tom70 led to a correction of Fis1 distribution in *mpf1Δ* cells with 70% mitochondrial and 25% peroxisomal Fis1, similar to the distribution observed in WT cells (Fig. 3B,C). Thus, it seems that the role of Mpf1 in regulating Fis1 distribution is dispensable in the presence of higher amounts of Tom70. Interestingly, only a partial correction of Fis1 distribution was observed upon overexpression of Tom70 in *tom71Δ* cells with 64% in mitochondria and 30% in peroxisomes (Fig. 3B,C), indicating that Tom71 has a unique role in Fis1 distribution that cannot be replaced by elevated levels of Tom70.

Remarkably, although we expected that overexpression of Tom70 in WT cells would drive Fis1 distribution more towards mitochondria, we observed a minor reduction in mitochondrial Fis1 (62%) and a slight increase in peroxisomal Fis1 (33%) (Fig. 3B,C). This observation is specially intriguing considering the report that the biogenesis of many mitochondrial proteins is enhanced upon overexpression of Tom70 (Liu et al, 2022). We can speculate that targeting of Fis1 towards mitochondria might prefer Tom71 over Tom70 and over-crowding the mitochondrial surface with Tom70 and/or engaging Tom71 in Tom71/Tom70 hetero-dimers creates a competing effect, thereby reducing Fis1 levels in mitochondria.

To further study the role of Tom71 in regulating the distribution of Fis1, we next constructed WT and *mpf1Δ* strains where Tom71 expression is under the control of the strong *GPD* promoter and could confirm the massive overexpression of Tom71 in these cells (Fig. 3D). Then, we separated mitochondria and peroxisomes from these strains and quantified the levels of mitochondrial and peroxisomal Fis1 (Fig. 3E,F). Interestingly, we observed that Tom71 over expression in both WT and *mpf1Δ* cells led to an increased distribution of Fis1 towards mitochondria (Fig. 3E,F). These findings substantiate the independent and unique contribution of Tom71 to the targeting of Fis1 to mitochondria.

Next, we wondered whether reduced or elevated levels of Tom70, Tom71, or Mpf1 influence either the steady-state levels of Fis1 or its stability. To that goal, we analyzed by Western blotting the amounts of Fis1 in mitochondria isolated from the relevant strains grown on oleate-containing medium. None of the tested strains displayed a significant alteration in the levels of Fis1 (Fig. EV3A). Then, we investigated the stability of Fis1 in the deletion or overexpressing cells by inhibiting translation of newly synthesized molecules of Fis1 with cycloheximide (CHX) and

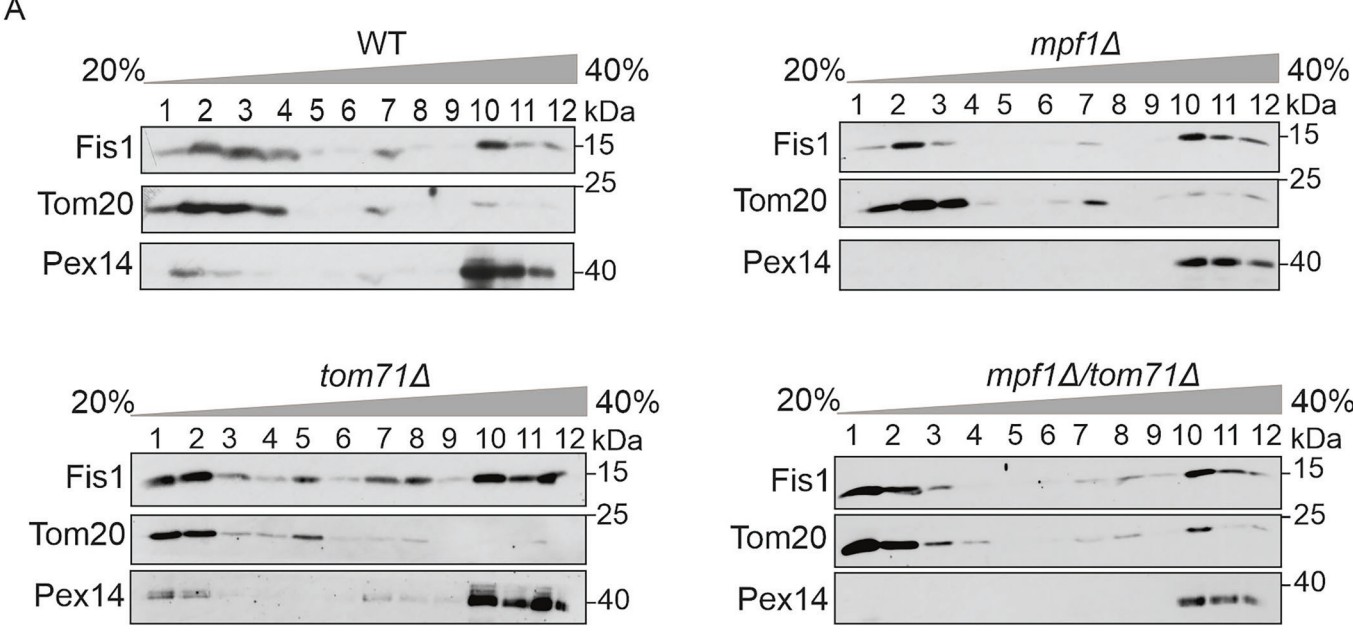

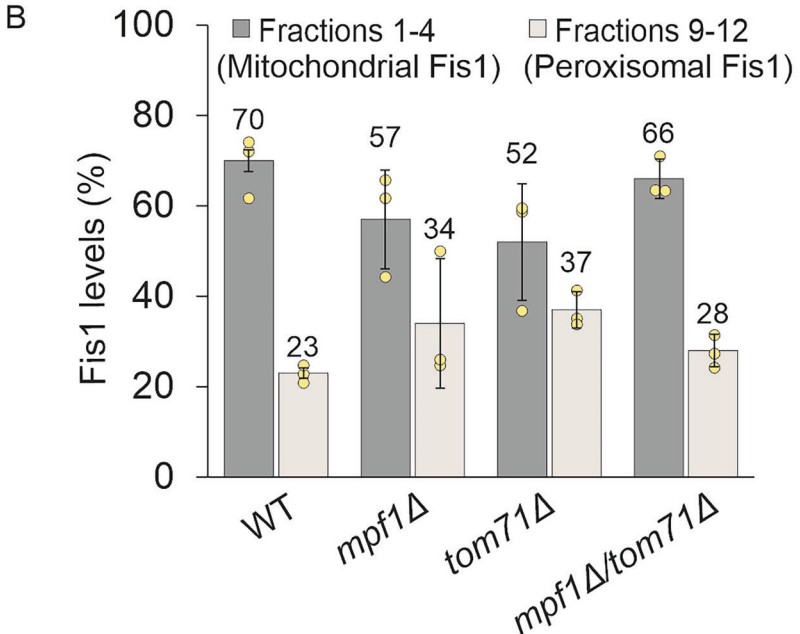

**Figure 2. Physical separation of mitochondria and peroxisomes validates the hits.**

(A) Gradient centrifugation procedure was employed to separate mitochondria and peroxisomes from the indicated strains and 12 fractions from the top of the gradient were collected. The fractions were analyzed by SDS-PAGE and immunodecoration with antibodies against Fis1 (dually localized to mitochondria and peroxisomes), Tom20 (mitochondrial marker), and Pex14 (peroxisome marker). (B) The intensities of Fis1 obtained in each fraction were quantified and the sum of all the 12 intensities was set to 100%. Fis1 signal in fractions 1–4 was considered mitochondrial, while that within fractions 9–12 was designated as peroxisomal. The graph represents the average of three independent experiments, error bars represent ±SD.

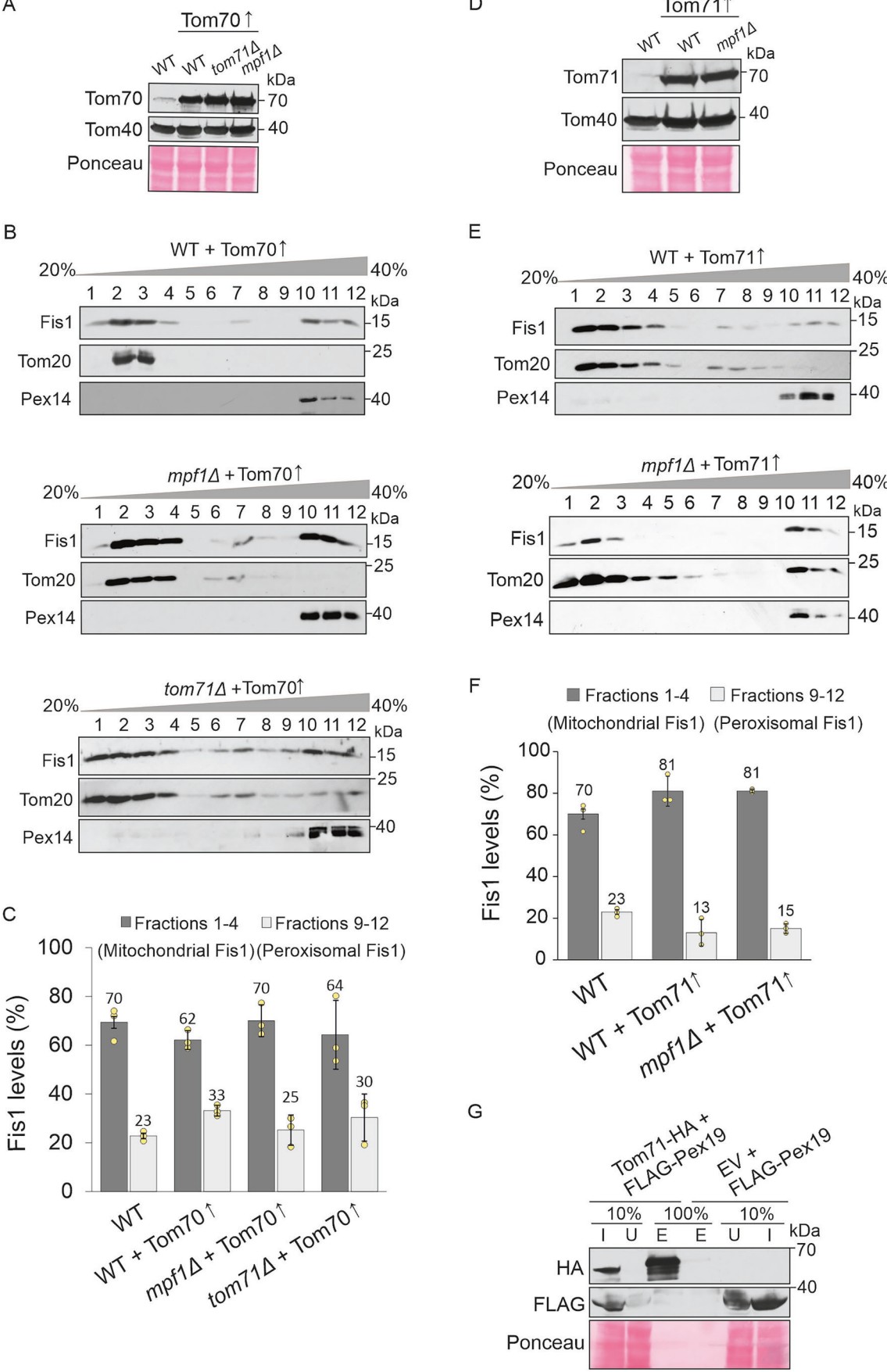

Figure 3.   Tom71 has a unique effect on the distribution of Fis1.

(A) Tom70 was overexpressed in the indicated strains by replacing the endogenous promoter with the *GPD* promoter. Cells of the resulting strains were grown on galactose and whole cell lysate was obtained by alkaline lysis. Extracted proteins were analyzed by SDS-PAGE and immunodecoration with the indicated antibodies. Ponceau staining was employed to verify equal loading in all lanes. (B, C) Gradient separation of mitochondria and peroxisomes from the indicated strains were performed and quantified as in Fig. 2A, B. The graph represents the average of three independent experiments, error bars represent ±SD. (D) Tom71 was overexpressed in the indicated strains by replacing the endogenous promoter with the *GPD* promoter. Proteins from the obtained strains were analyzed as described in part (A). (E, F) Gradient separation of mitochondria and peroxisomes from the indicated strains were performed and analyzed as in Fig. 2A, B. Note: to allow easier comparison, the Fis1 levels in WT cells in panels C and F were taken from Fig. 2B. The graph represents the average of three independent experiments, error bars represent ±SD. (G) Cells expressing either Flag-Pex19 alone or co-expressing Flag-Pex19 and Tom71-HA were lysed with Triton X-100 and the suspension was incubated with anti-HA beads. Fractions representing the input (I), unbound material (U), and the eluate (E) were analyzed by SDS-PAGE and immunodecoration with the indicated antibodies.

monitoring its overall levels three hours later. Consistent with the steady-state levels, also in this assay none of the strains with altered levels of Tom70, Tom71, or Mpf1 displayed changes in the stability of Fis1 (Fig. EV3B). Collectively, we conclude that the effect of the three proteins on the dual distribution of Fis1 is not due to their influence on its steady-state levels and/or stability.

It has previously been shown that the cytosolic chaperone/receptor Pex19 assists the biogenesis of mitochondrial Fis1 (Cichocki et al, 2018). However, because in the absence of Pex19 there are no peroxisomes at all, we did not consider *PEX19* as a hit in our initial visual screen. Since the overexpression of Tom71 drove 81% of Fis1 towards mitochondria, we wondered whether Tom71 functions as a receptor for Pex19 on the surface of mitochondria. To test this possibility, we created a strain where Tom71-HA and Flag-Pex19 are co-overexpressed in *tom70Δ* cells. We deleted Tom70 from these cells to prevent potential competition of Tom70 in binding to Pex19. Next, a pull-down analysis to detect potential interaction was performed. However, we could not co-elute Flag-Pex19 with Tom71-HA (Fig. 3G). This outcome proposes that the proteins might not interact, or that the interaction is very transient. Taken together, these results demonstrate a unique role of Tom71, whose absence cannot be compensated by Tom70, while the contribution of Mpf1 to the regulation of Fis1 distribution is dispensable upon overexpression of either Tom70 or Tom71.

## Deletion of *MPF1* is beneficial to cells grown on oleic acid

Since Mpf1 is an uncharacterized protein with an unknown function, we wanted to investigate whether loss of Mpf1 would influence growth of yeast cells. We noticed that *mpf1Δ* cells grew similar to WT cells when glucose (YPD), glycerol (YPG), or oleate (YPO) were used as carbon sources in a rich medium. However, when oleic acid was used as the sole carbon source on a synthetic medium, cells lacking Mpf1 grew better than WT cells (Fig. 4A).

In yeast, β-oxidation of fatty acids such as oleate takes place solely in peroxisomes (Hiltunen et al, 2003). Thus, yeast cells require fully functional peroxisomes for optimal growth on oleate as the exclusive carbon source. Previous studies have shown that absence of Fis1 reduces the number of peroxisomes in cells grown on oleate (Kuravi et al, 2006). Moreover, re-directing Fis1 only to peroxisomes by expressing Fis1-Pex15 fusion protein increased Dnm1-dependent peroxisome fission, and thereby increased the number of peroxisomes per cell (Motley et al, 2008). Hence, the better growth of *mpf1Δ* cells on oleate can be explained by the increased portion of Fis1 in peroxisomes in these cells, which in turn, enhances the number of peroxisomes and thus improves the utilization of oleate.

To test the feasibility of this scenario, we performed further experiments. First, to investigate the importance of Fis1 for growth on oleate, we compared the growth of cells lacking *FIS1* to that of control cells. Indeed, *fis1Δ* cells grown on oleate at elevated temperature (37 °C) displayed a clear growth retardation whereas these cells grew normally on glucose and exhibited only very minor growth phenotype on Glycerol, where fully functional mitochondria are required (Fig. 4B). Furthermore, we tested the number of peroxisomes in cells grown on either glucose or oleate and observed a clear increase on the latter carbon source (Fig. 4C). Of note, the average number of peroxisomes in cells grown on oleate was even further elevated upon the deletion of *MPF1* (Fig. 4C). Taken together, these findings support the proposal that the absence of Mpf1 is beneficial for cells grown on oleate due to the enhanced amounts of peroxisomal Fis1 under these conditions and the resulting elevated number of these organelles.

## Mpf1 is a highly unstable protein

A previous high throughput study proposed that Mpf1 might be a substrate of Grr1, an SCF ubiquitin ligase complex subunit. Both Mpf1 and its uncharacterized paralog, Yhr131c were suggested to interact with Grr1 and were reported to be partially stabilized in *grr1Δ* cells (Mark et al, 2014). To verify this previous report, we transformed a plasmid encoding Mpf1-3HA into WT and *grr1Δ* cells and monitored the life span of Mpf1 in these cells. In line with the previous findings, we observed that Mpf1 is indeed a highly unstable protein in WT cells and is almost completely degraded within 45 min after inhibition of translation by addition of cycloheximide (CHX) (Fig. 5A,B). However, in our hands, the deletion of *GRR1* had only a minor effect on the stability of Mpf1 (Fig. 5A,B), suggesting that there are other factors that affect the lifespan of this protein.

We next wished to determine the physiological relevance of the short life span of Mpf1. Considering the beneficial absence of Mpf1 on oleate, we investigated whether its steady-state levels are influenced by the carbon source. To that goal, we grew cells expressing Mpf1-HA on glucose and at a certain time point (Time = 0), we either shifted the cells to oleate-containing medium or kept them further on glucose. Interestingly, we observed extremely reduced amounts of Mpf1 after two hours on oleate (Fig. 5C). In contrast, the amounts of the control mitochondrial IMS protein, Tgl2-HA were not altered upon the shift to oleate (Fig. 5C). Of note, the transcript levels of *MPF1* on oleate as compared to glucose were not significantly reduced (Fig. 5D). These results suggest that the change in the protein amounts is related to its altered stability rather than changes in the mRNA levels. Thus, one can speculate

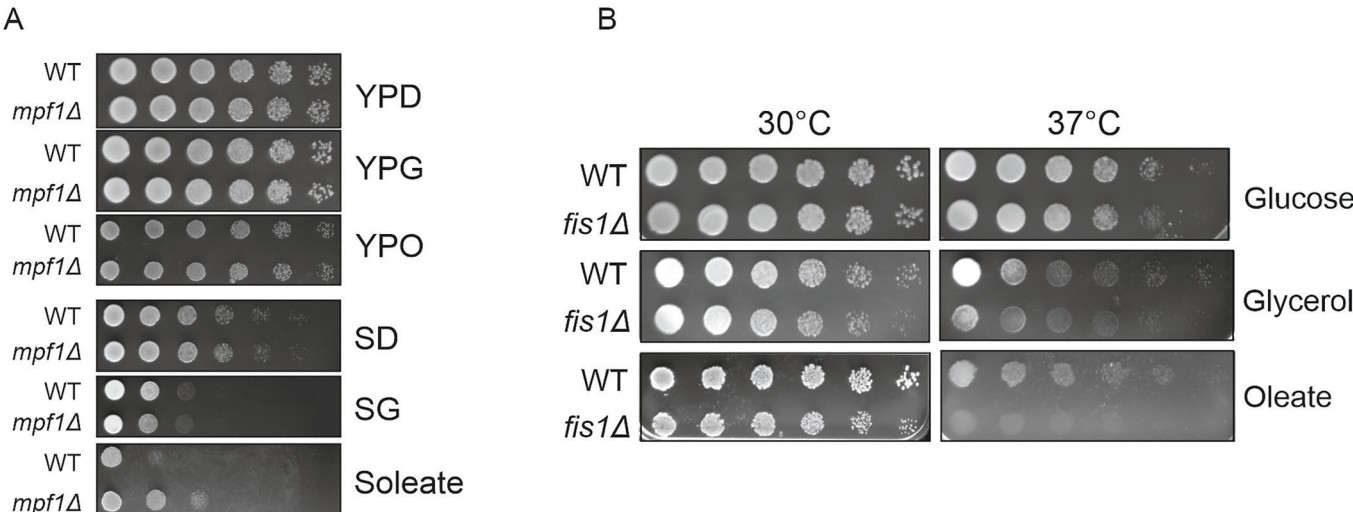

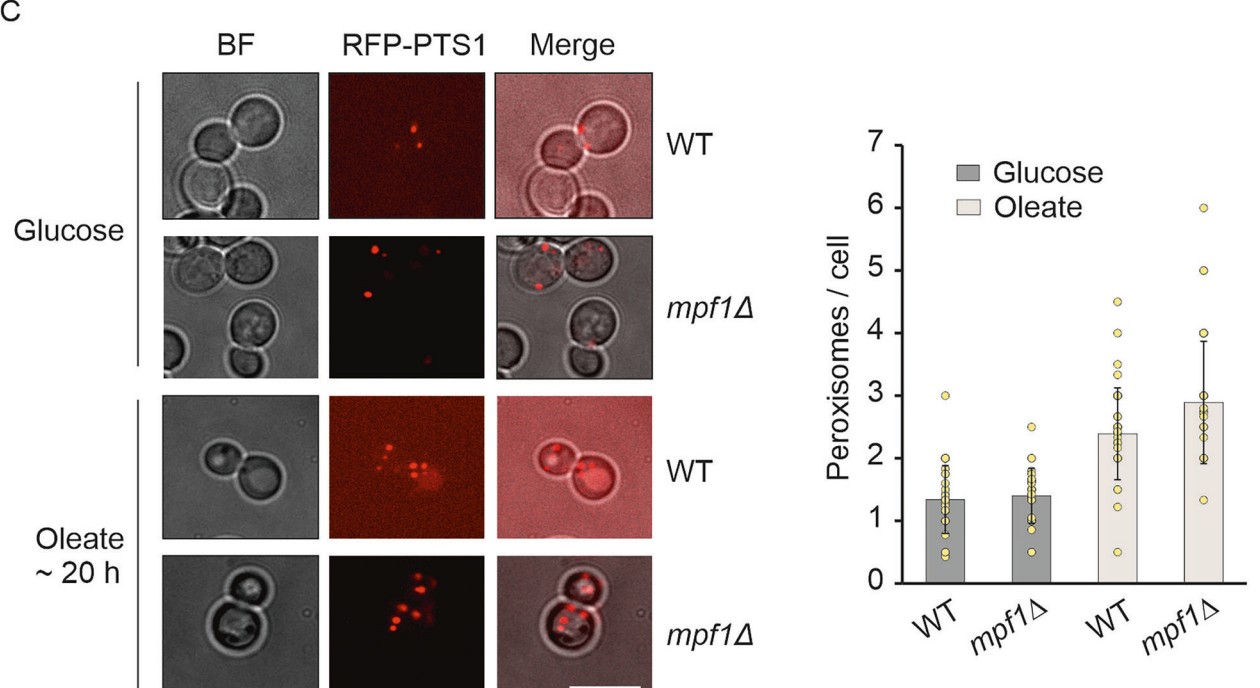

**Figure 4.   Loss of Mpf1 is beneficial for cells grown on oleic acid as a carbon source.**

(A) Growth of wild-type (WT) and *mpf1Δ* cells at 30 °C was analyzed by drop-dilution assay. The cells were grown on either rich media (YP) or synthetic media (S) containing glucose (YPD or SD), glycerol (YPG or SG), or oleate (YPO or SOleate). (B) Growth of wild-type (WT) and *fis1Δ* cells at either 30 °C or 37 °C was analyzed by drop-dilution assay. The cells were grown on media containing glucose, glycerol, or oleate as the sole carbon source. (C) Left panel: wild-type (WT) and *mpf1Δ* cells transformed with RFP-PTS1 to visualize peroxisomes were grown at 30 °C initially on glucose and then were either left on glucose or were shifted for 20 h to oleate containing medium. The images were taken in one focal plane. Scale bar, 5 μm. Right panel: the number of peroxisomes per cell was determined in cells grown under the conditions described for the left panel. At least 50 cells for each condition in three independent experiments were visualized. Error bars represent ±SD.

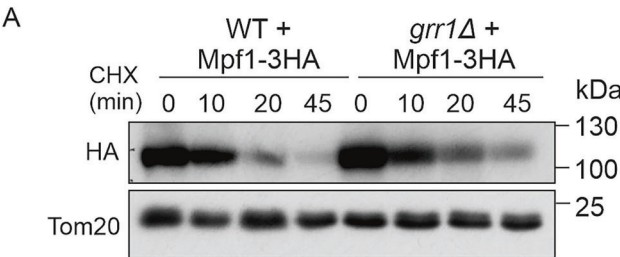

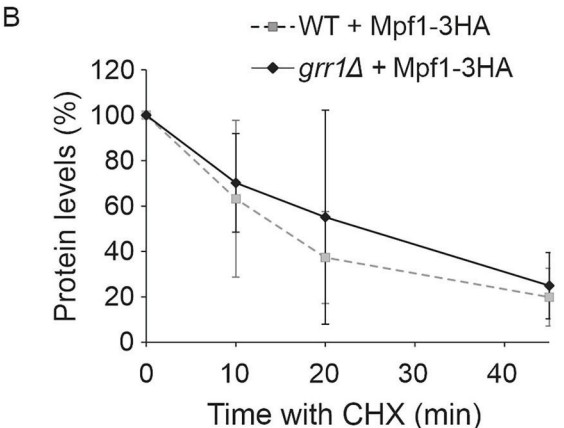

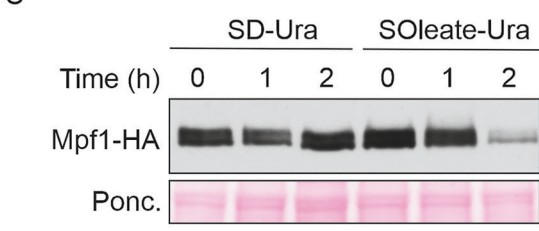

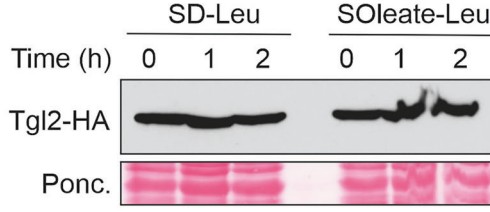

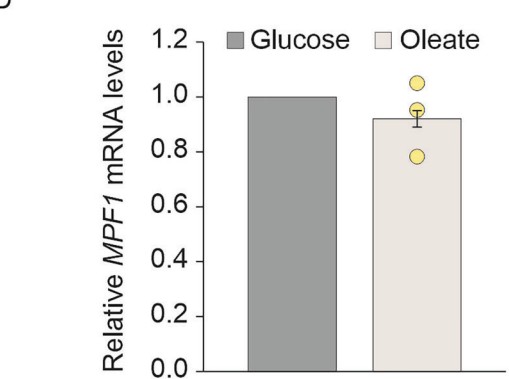

Figure 5. Mpf1 is an unstable protein.

(A) WT and *grr1Δ* cells were transformed with a vector encoding Mpf1-3HA. The cells were grown on SD-Ura and then at time = 0 the translation inhibitor cycloheximide (CHX) was added. Cells were further incubated, and proteins were extracted at each time point by alkaline lysis and analyzed by SDS-PAGE and immunodecoration with antibodies against either HA or Tom20 (as a loading control). (B) The bands representing either Mpf1-3HA or Tom20 were quantified and for each lane, the intensity of the band corresponding to Mpf1-3HA was normalized to the loading control (Tom20). The signal at time point = 0 was set to 100%. The average value for each time point of three independent experiments is shown. Error bars represent ±SD. (C) Cells harboring Mpf1-3HA (or Tgl2-HA as a control) were grown initially on synthetic medium containing glucose. Then, at time = 0 they were either left on glucose-containing medium or were shifted to oleate-containing medium. Samples were collected after the indicated time periods and proteins were extracted and analyzed by SDS-PAGE and immunodecoration with an anti-HA antibody. The Ponceau staining is shown as loading control. (D) To detect transcript levels of endogenous *MPF1*, RT-qPCR analysis was performed in WT cells overexpressing Mpf1-HA grown on either glucose or oleate as the sole carbon source. The transcript levels of ACT1 (encoding the abundant protein Actin) served as a reference and the amount of *MPF1* mRNA from cells grown on glucose was set as 1. N = 3, error bars represent ±SD.

that the rapid turnover of Mpf1 helps cells to better adapt to growth on oleate.

## Mpf1 loosely associates with the mitochondrial outer membrane

To further characterize Mpf1, we aimed to determine its subcellular location. WT cells expressing Mpf1-3HA were fractionated into whole cell lysate (WCL), ER, cytosol and mitochondria and these fractions were analyzed by Western blotting. Antibodies recognizing marker proteins for the mitochondria (Tom40), ER (Erv2), cytosol (Hexokinase), and peroxisomes (Pex14) were used to verify the fractionation. We detected Mpf1-3HA mainly in the mitochondrial fraction with some portion also in the ER fraction. Given the presence of the peroxisomal marker Pex14 in both fractions, we could only conclude that Mpf1-3HA might localize to mitochondria, peroxisomes, and/or the ER (Fig. 6A).

To obtain a more precise localization of Mpf1, we then chose to employ fluorescence microscopy techniques. Tagging Mpf1 with GFP led to cytosolic staining, which was not in line with the subcellular fractionation assays. This observation can be explained by either cleavage of the GFP tag and/or mis-targeting due to the bulky GFP moiety. Hence, we opted for immunofluorescence (IF) assays. Anti-HA antibodies conjugated to a fluorophore were used to detect Mpf1-3HA. As a control for the IF technique, we also visualized Tom22-HA, a bona fide mitochondrial outer membrane protein. We observed that Mpf1-3HA stained tubular structures that co-localize with RFP fused to mitochondrial targeting signal (RFP-MTS) but not with RFP-PTS1 (Peroxisomal targeting signal 1) (Fig. 6B). These findings support the notion that Mpf1-3HA localizes to mitochondria.

Since mitochondria have four different sub-compartments, we investigated the sub-mitochondrial localization of Mpf1-3HA by treating mitochondria isolated from cells expressing Mpf1-3HA with proteinase K (PK). We found Mpf1-3HA to be susceptible to PK digestion comparable to that of the surface proteins Tom70 and an Mcr1 isoform on the outer membrane (Mcr1$_{OM}$) (Fig. 6C). As expected for mitochondrial internal proteins, the Mcr1 isoform in the intramembrane space (Mcr1$_{IMS}$) and the matrix protein Hep1 were protected from PK by the outer membrane and both outer and

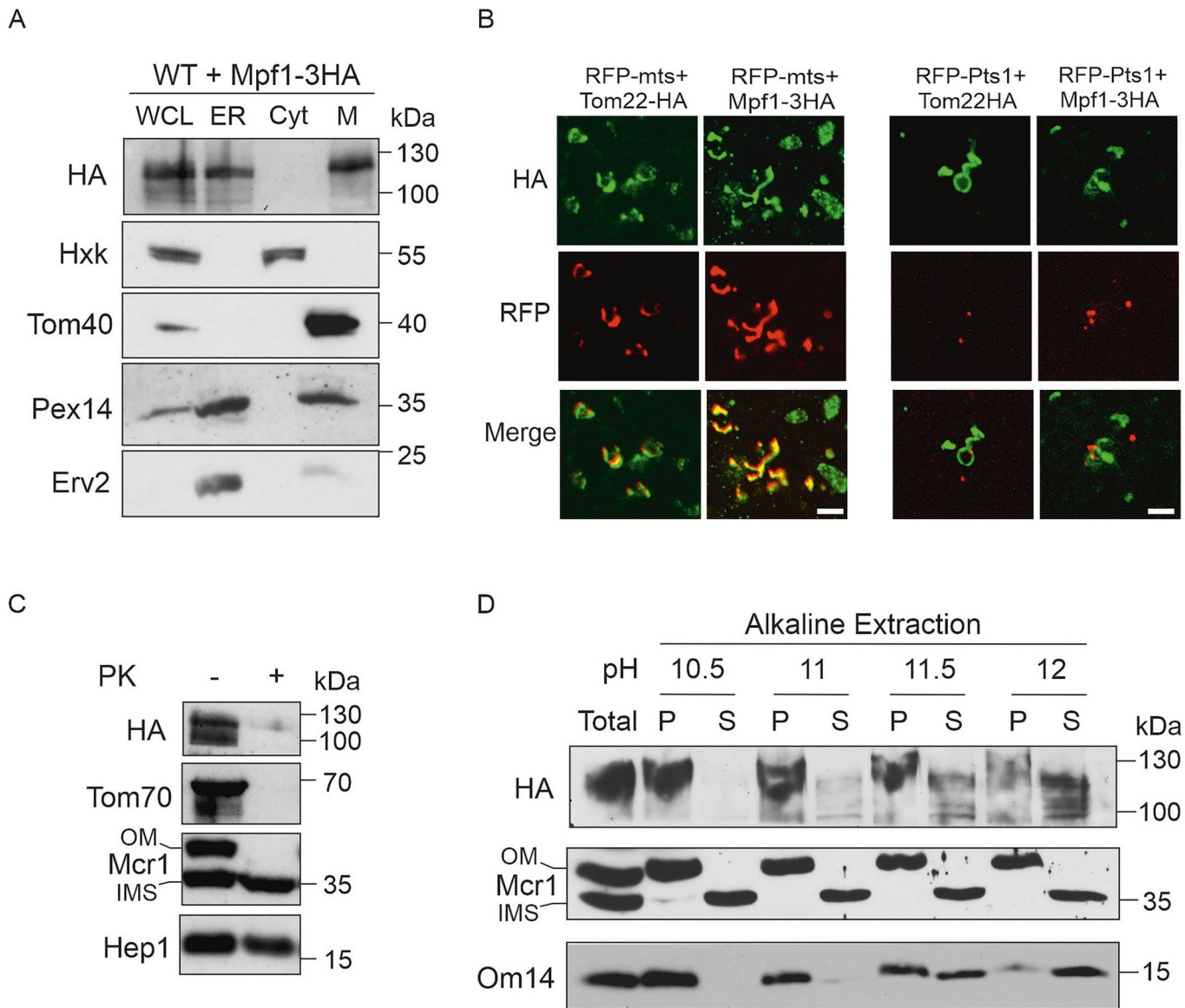

**Figure 6. Mpf1 shows a loose association with the mitochondrial outer membrane.**

(A) Cells overexpressing Mpf1-3HA were subjected to subcellular fractionation. The isolated fractions of whole cell lysate (WCL), microsomes (ER), cytosol (Cyt), and mitochondria (M) were analyzed by SDS-PAGE and immunodecoration with the indicated antibodies. Tom40 (mitochondria), Hexokinase (cytosol), Pex14 (peroxisomes), and Erv2 (ER) were used as marker proteins. (B) Cells expressing Mpf1-3HA were analyzed by immunofluorescence microscopy. *grr1Δ* cells were used to increase the lifespan of the protein. The HA tagged proteins were visualized with anti-HA antibody conjugated with Alexa Fluor™ 488. Tom22-HA, a bona fide mitochondrial protein was used as a control for the procedure. To visualize mitochondria and peroxisomes, the cells expressing the HA-tagged proteins were co-transformed with MTS-RFP (mitochondrial targeting signal) or RFP-PTS1 (peroxisomal targeting signal 1). Scale bar, 5 μm. (C) Isolated mitochondria from cells expressing Mpf-3HA were either left intact (−PK) or treated with proteinase K (+PK). Then, the samples were analyzed by SDS-PAGE and immunodecoration with the indicated antibodies. Tom70 and $Mcr1_{OM}$ are exposed on the mitochondrial surface whereas $Mcr1_{IMS}$ and Hep1 (matrix) are protected by mitochondrial membranes. (D) Isolated mitochondria from cells expressing Mpf-3HA were subjected to alkaline extraction using solution at the indicated pH values. "Total" represents untreated mitochondria. Membrane proteins were isolated in the pellet (P) fraction and soluble and membrane-peripheral proteins in the supernatant (S) fraction. The samples were analyzed by SDS-PAGE and immune-decoration against the specified antibodies. $Mcr1_{OM}$ and $Mcr1_{IMS}$ served as controls for integral membrane protein and soluble protein, respectively. Om14 acted as a control for MOM-associated protein extractable under extreme alkaline conditions.

inner membranes, respectively (Fig. 6C). Hence, we concluded that Mpf1 is exposed to the cytosol on the surface of mitochondria.

We next determined whether Mpf1-3HA is an integral or peripheral membrane protein by performing alkaline extraction of mitochondrial proteins followed by centrifugation to separate membrane embedded proteins in the pellet from soluble and

peripheral membrane proteins in the supernatant. To obtain a better resolution of the assay, we performed it under varying pH conditions. The alkaline pH decreases non-covalent protein-protein interactions and releases peripheral membrane proteins to the supernatant (Kim et al, 2015). As expected, the $Mcr1_{OM}$ isoform remained in the pellet fractions under all pH conditions,

confirming its behavior as a bona fide integral membrane protein. In contrast and as anticipated, the soluble IMS isoform of Mcr1 was in the supernatant fraction under all the employed conditions (Fig. 6D). We found that Mpf1-3HA was in the membrane fraction only in milder extraction condition (pH 10.5). As the pH was raised to 11, 11.5, and 12, increasing amounts of Mpf1 were found in the supernatant fraction. This behavior is similar to that of the outer membrane proteins Om14, which is known to be partially extractable under alkaline conditions (Burri et al, 2006; Zhou et al, 2022) (Fig. 6D). Altogether, these results indicate that Mpf1-3HA is peripherally associated with the cytosolic face of the mitochondrial OM.

## The combined loss of Tom70 and Tom71 affects the steady state levels of Mpf1

Considering the location of Mpf1 on the mitochondrial surface, we wondered whether Tom70, Tom71, or both are involved in targeting of Mpf1 to the organelle. Several studies on Tom70 and its paralog Tom71 (to a lesser extent) have identified a tetratricopeptide (TPR) structure that can bind to the cytosolic chaperones Hsp70 and Hsp90 and enable the recruitment of several precursor proteins together with chaperones to the mitochondrial surface (Backes et al, 2018; Jores et al, 2018; Young et al, 2003; Zanphorlin et al, 2016). Moreover, microscopy analysis of many GFP tagged proteins showed their reduced levels upon the absence of Tom70/71 (Backes et al, 2021). To investigate potential involvement of Tom70/71 in the biogenesis of Mpf1, we monitored the steady state levels of Mpf1-3HA in *tom70Δ*, *tom71Δ*, and double deletion *tom70Δ/71Δ* cells. Notably, in the absence of either Tom70 or Tom71 alone, Mpf1-3HA levels were comparable to those in WT cells. However, double deletion of both proteins resulted in a dramatic decrease compared to WT cells (Fig. 7A,B).

To understand the reason for such a striking reduction in the steady-state levels, we investigated whether the stability of Mpf1-3HA was affected in *tom70Δ/71Δ* cells. Though the relative levels of Mpf1-3HA were significantly lower in the double deletion cells at the beginning of the assay (time-point 0), the life span of the protein in the mutated cells was only moderately reduced as compared to WT cells (Fig. 7C,D). Furthermore, both subcellular fractionation and immunofluorescence microscopy revealed that Mpf1-3HA still localized predominantly to the mitochondria in *tom70Δ/71Δ* cells (Fig. 7E,F).

To better understand the potential involvement of Tom70 and Tom71 in the recruitment of Mpf1 to mitochondria, we created strains co-overexpressing Mpf1-3HA with either Tom70 or Tom71. Next, we performed subcellular fractionations of these cells and monitored the location of Mpf1. Our findings reveal that elevated levels of either Tom70 or Tom71 do not prevent the detection of a sub-population of Mpf1 in the ER/peroxisome fraction (Fig. EV4A,B). Hence, it appears that Tom70 and/or Tom71 cannot dictate alone the subcellular location of Mpf1.

Recent studies suggested that Tom70 can be involved in signaling through multiple transcription factors to control the transcription levels of genes encoding for many mitochondrial proteins. Accordingly, upon the selective removal of Tom70 from the mitochondrial surface, the levels of mRNAs encoding mitochondrial proteins were reduced (Liu et al, 2022). Along the same line, our RT-qPCR analysis revealed that the transcript levels

of endogenous *MPF1* as well as overexpressed *MPF1-3HA* were reduced by around 50% as compared to WT cells upon the deletion of both *TOM70/71* (Fig. EV5). These lower mRNA levels can explain (at least partially) the dramatically reduced levels of Mpf1 protein in the double deletion strain. It should be mentioned that such reduction in the detection of mRNA can result from less transcription of *MPF1*, increased degradation of the mRNA, and/or sequestering of the mRNA to P-bodies. Collectively, our findings suggest an involvement of both Tom70 and Tom71 in the transcriptional control of Mpf1-3HA. Although the mRNA and the protein steady-state levels of Mpf1-3HA are dramatically reduced in the absence of Tom70/71, Mpf1-3HA still localizes to mitochondria, suggesting the involvement of other factor(s) in its biogenesis.

## The PH domain of Mpf1 is involved in the stability of the protein but not in its localization

Large scale studies and structural analysis predicted the presence of a Pleckstrin homology (PH) domain in Mpf1 (Gallego et al, 2010; Isakoff et al, 1998; Lemmon, 2004). This domain can interact with phosphoinositide (PI) species as well as other lipids on biological membranes. Previous attempts to investigate binding of a recombinant PH domain from Mpf1 to PI were unsuccessful due to inadequate quantities (Yu et al, 2004). PH domains occur in a wide range of proteins with varying functions, and are stretches of ~120 amino acid residues with two anti-parallel β-sheets followed by a C-terminal α-helix (Riddihough, 1994). Membrane targeting of PH domains that strongly bind to PIs could be abolished by mutating the basic residues in the β1/β2 loop (Yu et al, 2004). To investigate if the PH domain of Mpf1 is essential for the stability and/or subcellular localization of the protein, we created a mutant of Mpf1 with lysine and arginine residues in the PH domain replaced by alanine (K144A, K147A, and R157A) (Fig. 8A, Mpf1(PH*)-3HA). Compared to regular Mpf1-3HA, Mpf1(PH*)-3HA exhibited elevated stability, with steady-state levels ~60% higher than those of the native protein (Fig. 8B,C). The PH domain is known to serve as a platform for protein-protein interactions (Lemmon, 2004; Scheffzek and Welti, 2012), and mutating the basic residues in the PH domain of Mpf1 might have disrupted its interactions with some factors of the proteasome degradation pathway and thereby increasing its stability.

Of note, cells overexpressing Mpf1(PH*)-3HA grew slightly better than WT cells on oleate-containing medium (Fig. 8D). It can be speculated that this variant might have a dominant negative effect on the recruitment of Fis1 to mitochondria and thus enhance the number of peroxisomes under these conditions.

We next asked whether the basic residues in the PH domain of Mpf1 are required for its targeting to mitochondria and its association with the OM. Employing immunofluorescence microscopy and subcellular fractionation, we found that Mpf1(PH*)-3HA still localizes to the mitochondria (Fig. 9A,B). Molecular modeling revealed that the PH domain of Mpf1 has an overall weak or no positive charge, and membrane targeting or binding to PIs may depend on strong positive charges (Yu et al, 2004). Furthermore, not all PH domains exhibit strong and specific interactions with PIs and many of them bind to PIs with low affinity and specificity. Effective binding of such "weak" PH domains to biological membranes might be strengthened by interactions with other membrane-bound proteins (Maffucci and Falasca,

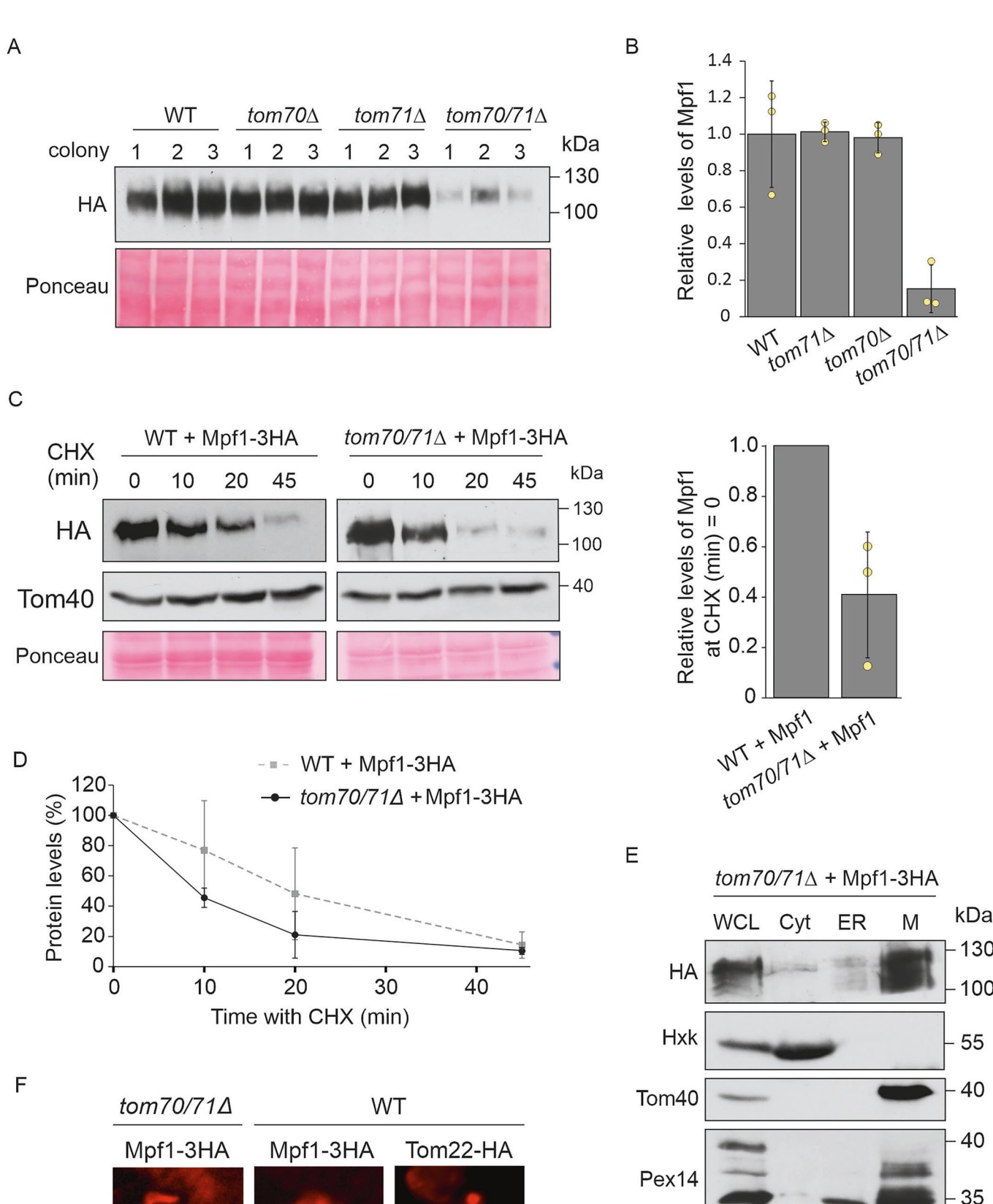

Figure 7. The absence of Tom70 and Tom71 reduces the expression of Mpf1.

(A) Proteins were extracted from the indicated cells (three independent colonies that were selected after transformation with the Mpf1-HA encoding plasmid)) expressing Mpf1-3HA. Samples were analyzed by immunodecoration with HA antibody. Ponceau staining was used as the loading control. (B) The bands representing Mpf1-3HA in all the analyzed cells were quantified and for each lane normalized to the intensity of the corresponding Ponceau staining. For each strain, the average of the three colonies analyzed in (A) was calculated. The average value for WT cells was set as 1. $N = 3$. Error bars represent ±SD. (C) Left panel: WT and tom70/71Δ cells expressing Mpf1-3HA were subjected to cycloheximide (CHX) assay as described in the legend to Fig. 5A. Right panel: Quantification of Mpf1-HA levels relative to Ponceau staining at time point = 0. $N = 3$. (D) The bands corresponding to Mpf1-3HA in the experiment presented in (C) were quantified as described in the legend to Fig. 5B. The average value for each time point in three independent experiments is shown. Error bars represent ±SD. (E) Subcellular fractionation of tom70/71Δ cells expressing Mpf1-3HA. The isolated fractions of whole cell lysate (WCL), microsomes (ER), cytosol (Cyt), and mitochondria (M) were analyzed by SDS-PAGE and immunodecoration with the indicated antibodies. Tom40 (mitochondria), Hexokinase (cytosol), Pex14 (peroxisomes), and Erv2 (ER) were used as marker proteins. (F) Immunofluorescence microscopy to visualize Mpf1-3HA in tom70/71Δ and WT cells. Tom22-HA was used as a control for the IF procedure. The HA-tagged proteins were visualized using an anti-HA antibody conjugated with Alexa Fluor™ 594. Scale bar, 5 μm.

2001). To explore this option, we then performed an alkaline extraction to see if the association of Mpf1(PH*)-3HA with the MOM is affected. Mpf1(PH*)-3HA predominantly remained in the pellet fraction in pH 10.5 and there was an increase of the protein portion in the supernatant only at pH 12, indicating that Mpf1(PH*)-3HA exhibits a similar, or even somewhat stronger association to the MOM than its native counterpart (Fig. 9C). Collectively, these results suggest that mutating key residues in the PH domain of Mpf1 did not change its association with mitochondria.

## Discussion

Considering the number of shared proteins between mitochondria and peroxisomes and the de novo formation of peroxisomes via mitochondria-derived vesicles (at least in mammalian cells), several studies speculate that peroxisomes evolved to facilitate the quality control of mitochondria under periods of stress and to relieve mitochondria from the burden of hosting oxidation enzymes of the β-oxidation pathway (Bittner et al, 2022; Speijer, 2017). This symbiotic relationship between mitochondria and peroxisomes might have allowed peroxisomes to utilize several mitochondrial proteins for their own needs like those proteins involved in fission (Fis1) and quality control (Msp1). Moreover, mitochondria contribute indirectly to the targeting of the phosphatase Ptc5p to peroxisomes via a mitochondrial transit. These cross-talks are proposed to eventually lead to targeting of these proteins to both organelles when peroxisomes became autonomous over time (Bittner et al, 2022; Stehlik et al, 2020). Distribution of the dually localized TA protein Fis1 to mitochondria and peroxisomes is aided by Pex19 (Cichocki et al, 2018). However, regulatory mechanisms to understand how the distribution of these cytosolically synthesized TA proteins is controlled have remained elusive.

In this study, we used comprehensive techniques to find factors that regulate the dual distribution of the TA proteins Fis1 and Gem1 to the membranes of mitochondria and peroxisomes. Using high-throughput microscopy screens, we first identified the involvement of Tom70, Tom71, and an uncharacterized protein Ynl144c (re-named as Mpf1 in this study) in regulating the distribution of Fis1 and Gem1 to mitochondria and peroxisomes. Subsequently, we verified the involvement of these candidates via subcellular fractionation assays and confirmed that in tom71Δ and mpf1Δ cells, Fis1 distributed more to peroxisomes. Surprisingly, the double deletion cells tom71Δ /mpf1Δ showed Fis1 distribution comparable to WT cells. Since mitochondrial fission is crucial for

the maintenance of healthy cells and the depletion of Fis1 leads to hyperfused mitochondria (Das and Chakrabarti, 2020; Hoppins et al, 2007), tom71/mpf1Δ potentially leads to activation of other factors to maintain proper distribution of Fis1 molecules. Further investigations on which factors could be upregulated in tom71/mpf1Δ cells would enhance our understanding of this regulatory mechanism. Unfortunately, due to technical difficulties, we were unable to optimally separate mitochondria and peroxisomes in tom70Δ and tom70/71Δ cells, and thereby the impact of the absence of Tom70 on the distribution of Fis1 could not be verified.

However, we were able to show that enhanced levels of Tom70 in mpf1Δ cells could fully correct the Fis1 distribution to WT levels but only partially reverse the distribution in tom71Δ. This observation suggests that Tom71 might have a more dominant role in affecting Fis1 distribution, that is not entirely rectified by higher levels of Tom70. Although Tom71 was initially identified more than 25 years ago as a barely expressed paralog of Tom70 (Schlossmann et al, 1996), until this study, a distinct function of Tom71 distinguishing it from Tom70 was not found. In contrast, the more abundant paralogue Tom70 was reported to fulfill several functions like import receptor (Brix et al, 1999; Yamamoto et al, 2009), docking site for cytosolic chaperones (Backes et al, 2021; Kreimendahl and Rassow, 2020; Young et al, 2003; Jores et al, 2018; Drwesh et al, 2022), tether protein in mitochondria contact sites (Eisenberg-Bord et al, 2021; Koch et al, 2024), and transcription regulator (Liu et al, 2022). Our hypothesis that Tom71 plays a unique role in regulating Fis1 distribution is further validated by our finding that overexpression of Tom71 has a significant effect (compared to that of Tom70) in driving Fis1 distribution more towards mitochondria. A possible explanation for the effect of Tom71 is a putative role as a mitochondrial receptor for Pex19. However, so far, we have not detected physical interaction between Tom71 and Pex19. It could be that the interaction is rather transient and requires the presence of substrate protein in transit.

Since Tom70/71 are known to function as receptors for newly synthesized mitochondrial proteins as well as docking site for Hsp70 and Hsp90 chaperones, the involvement of these paralogues can be on both levels. Tom70/71 can associate with the TMD of TA proteins and/or they can interact with the chaperones that join the TA proteins on their journey to mitochondria. This latter scenario is supported by our previous finding that the cytosolic Hsp70 (Ssa1) and its co-chaperone Sti1 contribute to the biogenesis of mitochondrial TA proteins (Cichocki et al, 2018). The fact that the distribution of Fis1 in mpf1Δ cells could be completely corrected by higher levels of either Tom70 or Tom71 shows that

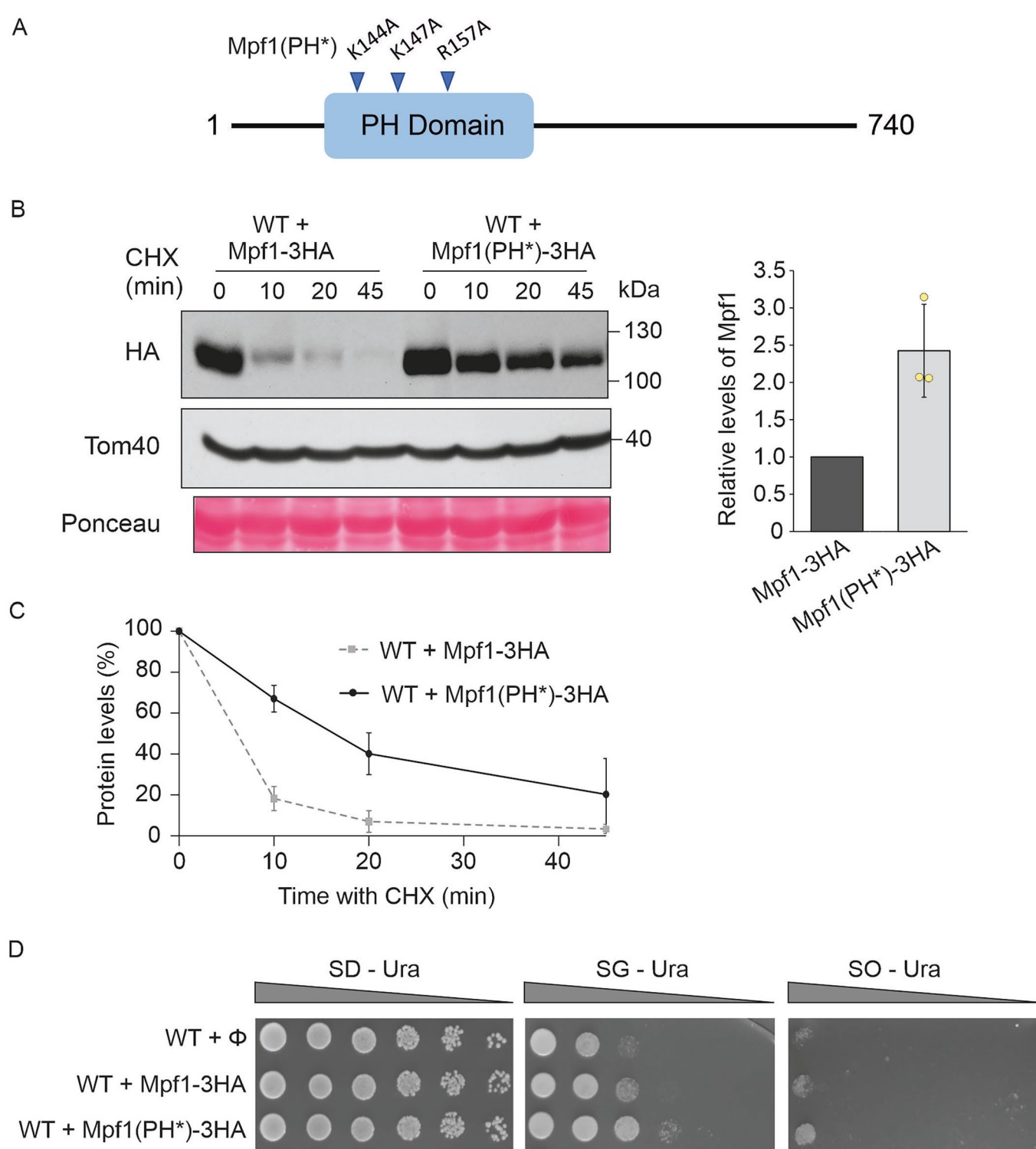

the role of Mpf1 in regulating Fis1 is dispensable upon over-expression of one of these paralogues. This complementation can also suggest that Mpf1, Tom70, and Tom71 potentially share the same pathway in regulating the trafficking of Fis1 to mitochondria and peroxisomes. Our experiments to test whether Mpf1 directly interacts with Tom70 and/or Tom71 could not provide support for such interaction (Fig. EV6).

Although the precise molecular function of Mpf1 is not known yet, a hint for its physiological role is provided by our finding that yeast cells grown on fatty acid (oleate) benefit from the deletion of *MPF1*. Moreover, *mpf1Δ* cells grown on oleate have more peroxisomes than control cells under these conditions. We assume that this phenotype is due to an increase in the quantity of Fis1 molecules targeted towards peroxisomes, which subsequently enhance the division of peroxisomes

Figure 8. Mutating the Pleckstrin homology (PH) domain of Mpf1 stabilizes the protein.

(A) Schematic diagram of the mutations in the PH domain of Mpf1 with K144, K147, and R157 replaced by alanine (A) residues (the mutant is indicated as Mpf1(PH*). The mutated basic residues in the in the β1/β2 loop of the PH domain are indicated with blue arrowheads. (B) Left panel: WT cells overexpressing either Mpf1-3HA or Mpf1(PH*)-3HA were subjected to cycloheximide (CHX) assay for the indicated time periods. Proteins were then extracted and analyzed by SDS-PAGE and immunodecoration with antibodies against either HA or Tom40 (as loading control). Right panel: Quantification of Mpf1-HA and Mpf1(PH*)-3HA levels relative to Tom40 (as loading control) at time point = 0. The average of three independent experiments is shown. The levels of native Mpf1-HA were set as 1. N = 3. Error bars represent ±SD. (C) Quantification of Mpf1-3HA and Mpf1(PH*)-3HA was performed as described in Fig. 5B. The average value for each time point of three independent experiments is shown. Error bars represent ±SD. (D) Growth analysis by drop dilution assay of WT cells harboring either an empty vector (Φ), a plasmid encoding Mpf1-3HA, or a plasmid encoding Mpf1(PH*)-3HA. Cells were grown at 30 °C on synthetic media containing glucose (SD-Ura), glycerol (SG-Ura), or oleic acid (SO-Ura).

A

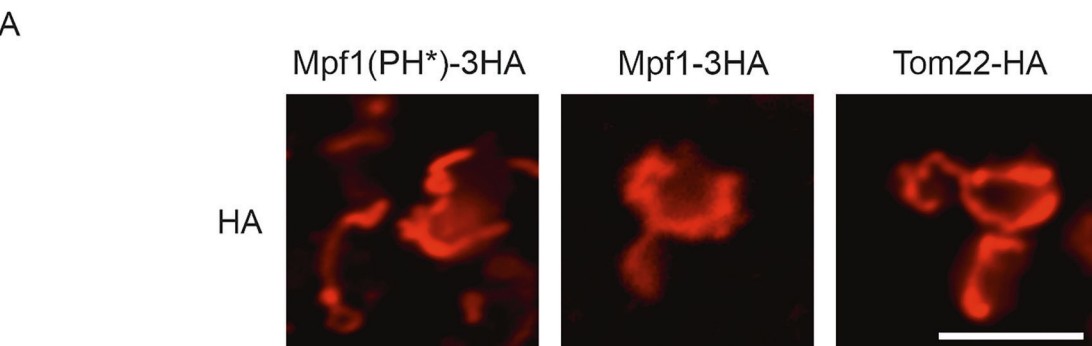

B

C

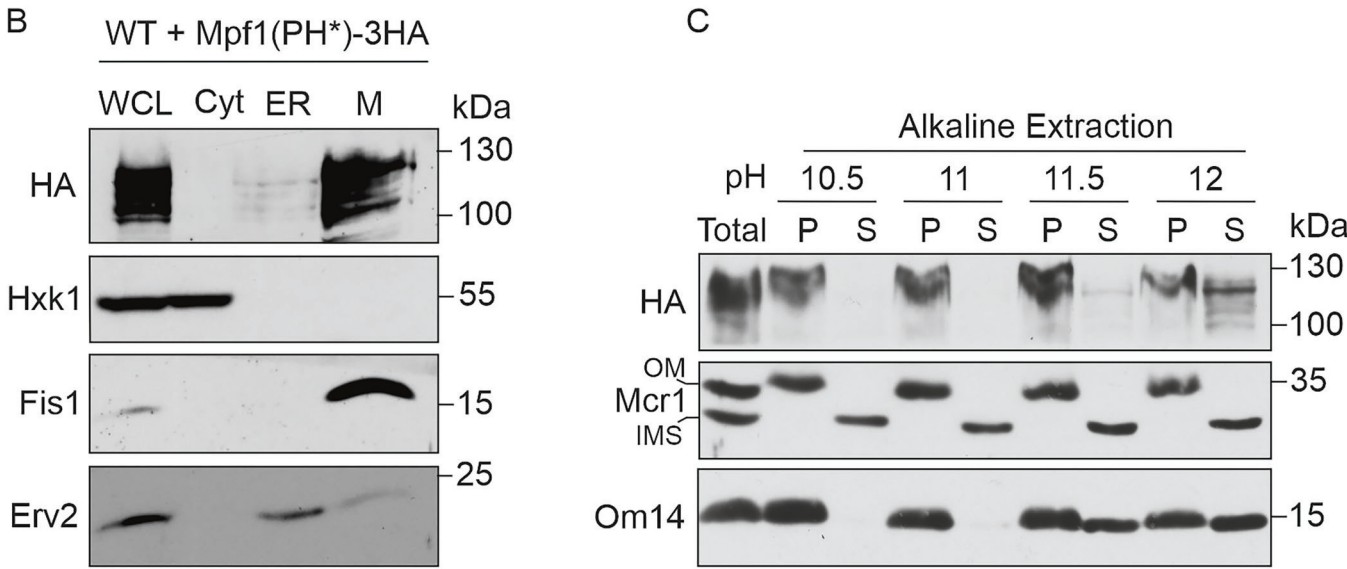

**Figure 9. The mutations in the PH domain do not affect the location of Mpf1.**

(A) Immunofluorescence microscopy localization of Mpf1(PH*)-3HA in WT cells. Native Mpf1-3HA and Tom22-HA were used as a control for the procedure. The HA-tagged proteins were visualized using an anti-HA antibody conjugated with Alexa Fluor™ 594. Scale bar, 5 μm. (B) Subcellular fractionation of WT cells expressing Mpf1(PH*)-3HA. Cells were analyzed as described in the legend to Fig. 6A. (C) Isolated mitochondria from cells expressing Mpf1(PH*)-3HA were subjected to alkaline extraction as described in Fig. 6D.

and thus their number, which could be particularly beneficial when oleate is the sole carbon source. Our observation that cells lacking Fis1 show growth retardation on oleate further supports the importance of Fis1 for the performance of peroxisomes. However, it might also be that in the absence of Mpf1 other cellular mechanisms

(beside increased peroxisomal Fis1 levels) cause beneficial effects for growth in oleate media.

Another interesting aspect of Mpf1 is its rather short lifespan. This inherent instability raises the question why cells produce protein molecules that will be degraded within minutes. The major

reduction in the levels of Mpf1 upon shifting the carbon source to oleate suggests that such a short life span might be beneficial for the adaptation of cells to oleate as a carbon source. Currently, we can only speculate that under some other rather special conditions the presence of Mpf1 could be required immediately and obtaining new molecules via enhancing transcription and translation might be too time-consuming. In the future, it will be of interest to identify conditions that support enhanced stability of Mpf1.

Characterizing the subcellular localization of Mpf1 indicated it to be a peripheral membrane protein, loosely associated to the mitochondrial OM. Interestingly, even in the absence of both Tom70 and Tom71, Mpf1 still makes its way to the mitochondrial surface. Along the same line, overexpression of either Tom70 or Tom71 does not change the subcellular distribution of Mpf1. These observations suggest the involvement of other factors that can mediate the association of Mpf1 to the organelle. Mutating the conserved basic residues in the β1/β2 loop and thereby potentially disrupting the PH domain of Mpf1 did not hamper its mitochondrial localization either. Thus, it seems that either the triple mutation did not interfere completely with the function of the PH domain or other regions of the protein facilitate the association with mitochondria. Our finding that Mpf1(PH*) exhibited enhanced stability might indicate that certain proteins in the Ubiquitin/proteasome degradation pathway usually recognize a degron element in the PH domain but are unable to do so with the mutated PH domain. Alternatively, it might be that the mutations stabilize the interaction of Mpf1 with a protein and/or a lipid in the mitochondria outer membrane and through these interactions Mpf1 is stabilized.

Altogether, our findings contribute to novel insights on factors responsible for regulating the dual distribution of Fis1 to mitochondria and peroxisomes. We identified for the first time three proteins Tom70, its paralogue Tom71, and Mpf1 as involved in this process. In addition to recognizing a unique function of Tom71, we could provide a function for a so far uncharacterized protein—Mpf1. We identify the latter as an unstable protein at the surface of mitochondria that its absence is beneficial for cells grown on oleate and in need for more peroxisomes. Collectively, the current study provides the first glimpse into the process of dual distribution of TA proteins between mitochondria and peroxisomes.

# Methods

## Yeast strains and growth conditions

*Saccharomyces cerevisiae* strains used in this study are listed in Reagent and Tools Table. For induction of peroxisomes, yeast strains were grown at 30 °C on oleate-containing YNBO media (0.1% (w/v) yeast extract, 0.17% (w/v) yeast nitrogen base, 0.5% (w/v) ammonium sulfate, 0.0002% (w/v) uracil, 0.0002% (w/v) adenine sulfate, 0.12% oleic acid, 0.2% Tween40, supplied with amino acids). Generally, cells were grown at 30 °C on selective or rich media (YP) supplemented with 2% of either glucose or galactose. Yeast transformation was performed by the lithium acetate method (Gietz and Woods, 2006).

## Yeast growth assay

Yeast strains were cultivated till mid-logarithmic phase and after harvesting them, cells were resuspended to 1 ml of $OD_{600} = 2$. The

cell suspension was fivefold serially diluted and 5 µl of each dilution was spotted on the indicated solid media. The plates were incubated at 30 °C in a humid box and the growth was monitored for 2–10 days.

## High-throughput microscopy screening

The following query strains were made to cross with the yeast deletion library: (i) mCherry-Fis1, Pex3-GFP, (ii) mCherry-Fis1, Om45-GFP, (iii) mCherry-Gem1, Pex3-GFP, and (iv) mCherry-Gem1, Om45-GFP. To generate these query strains a DNA sequence encoding the mCherry tag was genomically inserted by homologous recombination at the 5' of the sequence encoding either Fis1 or Gem1, with the strong and constitutive *TEF2* promoter and the Nourseothricin N-acetyl transferase (NAT) selection cassette. Subsequently, in these strains the DNA sequence encoding GFP was integrated by homologous recombination into the 3' region of either Pex3 or Om45 with the *TEF2* promoter and the hygromycin B phosphotransferase (HPH) selection cassette. Query strains were crossed by synthetic genetic array (SGA) with two libraries—the knock out (KO) library and the decreased abundance by mRNA perturbation (DAmP) library, as previously described (Cohen and Schuldiner, 2011; Tong and Boone, 2006). The high-throughput screen was performed by growing cells overnight at 30 °C in rich media (YP) supplemented with galactose, diluting them 1:10 in the next morning and letting them divide at 30 °C for 4 h before imaging with an automated system (Breker et al, 2013).

## Recombinant DNA techniques

Full lists of primers and plasmids used in this study are found in Reagent and Tools Table. The plasmid pGEM4-Mpf1-3HA was used as a template for site directed mutagenesis to create the PH domain mutant of Mpf1. The PCR product was digested with Dpn1 and transformed into *E. coli* cells. For gene deletion and manipulation, PCR product containing the selection cassette with flanking regions complementary to DNA sequences of the gene of interest was transformed into yeast cells by the Li-acetate method. Colonies were analyzed by screening PCR. All constructs were verified by DNA sequencing.

## Separation of mitochondria and peroxisomes by gradient centrifugation

A detailed description of this separation procedure can be found in a recently published article (Aravindan and Rapaport, 2024). Shortly, yeast cells were precultured overnight in 100 ml YP medium supplemented with 0.1% glucose. Next morning, the culture was upscaled to 400 ml and incubated overnight. For induction of peroxisomes, cells were harvested (5000 × *g*, 6 min, RT) and washed with 20 ml sterile water and centrifuged again (5000 × *g*, 6 min, RT). Cells were then resuspended in 1000 ml YNBO medium and incubated for 16–20 h. Then, cells were harvested (5000 × *g*, 6 min, RT) and washed twice with 30 ml sterile water followed by centrifugation (5000 × *g*, 6 min, RT). The cells were then incubated in 20 ml Dithiothreitol (DTT) buffer (100 mM Tris, 10 mM DTT) for 30 min and were harvested (1500 × *g*, 6 min, RT) and washed twice with 20 ml sorbitol buffer (20 mM

**Reagents and tools table**

| Reagent/Resource | Reference or Source | Identifier or Catalog Number |
|---|---|---|
| **Experimental models** | | |
| WT (BY4741) MATa his3Δ1 leu2Δ0 met15Δ0 ura3Δ0 | Lab stock | YDR4845 |
| TEF2 mCherry *FIS1* (BY4741) MATa his3Δ1 leu2Δ0 met15Δ0 ura3Δ0 lys+ can1Δ::GAL1pr-SceI::STE2pr-SpHIS5 lyp1Δ::STE3pr-LEU2 NAT::TEF2pr-mCherry-Fis1 | Lab of Maya Schuldiner | |
| TEF2-mCherry *GEM1* (BY4741) MATa his3Δ1 leu2Δ0 met15Δ0 ura3Δ0 lys+ can1Δ::GAL1pr-SceI::STE2pr-SpHIS5 lyp1Δ::STE3pr-LEU2 NAT::TEF2pr-mCherry-Gem1 | Lab of Maya Schuldiner | |
| TEF2 mCherry *FIS1*, *OM45* GFP (BY4741) MATa his3Δ1 leu2Δ0 met15Δ0 ura3Δ0 lys+ can1Δ::GAL1pr-SceI::STE2pr-SpHIS5 lyp1Δ::STE3pr-LEU2 NAT::TEF2pr-mCherry-Fis1 OM45-GFP::hph | This study | YDR4528 |
| TEF2 mCherry *FIS1*, *PEX3* GFP (BY4741) MATa his3Δ1 leu2Δ0 met15Δ0 ura3Δ0 lys+ can1Δ::GAL1pr-SceI::STE2pr-SpHIS5 lyp1Δ::STE3pr-LEU2 NAT::TEF2pr-mCherry-Fis1 PEX3-GFP::hph | This study | YDR4530 |
| TEF2-mCherry *GEM1*, *OM45* GFP (BY4741) MATa his3Δ1 leu2Δ0 met15Δ0 ura3Δ0 lys+ can1Δ::GAL1pr-SceI::STE2pr-SpHIS5 lyp1Δ::STE3pr-LEU2 NAT::TEF2pr-mCherry-Gem1 OM45-GFP::hph | This study | YDR4532 |
| TEF2-mCherry *GEM1, PEX3* GFP (BY4741) MATa his3Δ1 leu2Δ0 met15Δ0 ura3Δ0 lys+ can1Δ::GAL1pr-SceI::STE2pr-SpHIS5, lyp1Δ::STE3pr-LEU2 NAT::TEF2pr-mCherry-Gem1 PEX3-GFP::hph | This study | YDR4534 |
| *mpf1*Δ (BY4741) MATa, his3Δ1, leu2Δ0, met15Δ0, ura3Δ0, YNL144C::KAN | Euroscarf | YDR4848 |
| *tom71*Δ (BY4741) MATa, his3Δ1, leu2Δ0, met15Δ0, ura3Δ0, YHR117W::KAN | Euroscarf | YDR4844 |
| *tom70*Δ (BY4741) MATa, his3Δ1, leu2Δ0, met15Δ0, ura3Δ0, YNL121C::KAN | This study | YDR2617 |
| *tom70/71*Δ (BY4741) MATa his3Δ1 leu2Δ0 met15Δ0 ura3Δ0 TOM70::KAN, TOM71::Nat2 | This study | YDR4847 |
| *mpf1/tom71*Δ (BY4741) MATa his3Δ1 leu2Δ0 met15Δ0 ura3Δ0 YNL144C::KAN, HR117W::NAT | This study | YDR4790 |
| GPD Tom70 (BY4741) MATa his3Δ1 leu2Δ0 met15Δ0 ura3Δ0 NAT::GPDpr TOM70 | This study | YDR4838 |
| *tom71*Δ, GPD *TOM70* (BY4741) MATa his3Δ1 leu2Δ0 met15Δ0 ura3Δ0 YHR117W::NAT NAT::GPDpr TOM70 | This study | YDR4811 |
| *mpf1*Δ, GPD *TOM70* (BY4741) MATa his3Δ1 leu2Δ0 met15Δ0 ura3Δ0 YNL144C::KAN, NAT::GPDpr TOM70 | This study | YDR4837 |
| GPD *TOM71* (BY4741) MATa his3Δ1 leu2Δ0 met15Δ0 ura3Δ0 NAT::GPDpr TOM71 | This study | YDR4839 |
| *mpf1*Δ, GPD *TOM71* (BY4741) MATa his3Δ1 leu2Δ0 met15Δ0 ura3Δ0 YNL144C::KAN, NAT::GPDpr TOM71 | This study | YDR4836 |
| *grr1*Δ (BY4741) MATa his3Δ1 leu2Δ0 met15Δ0 ura3Δ0 GRR1::kanMX4 | Euroscarf | YDR4830 |
| **Recombinant DNA** | | |
| pYMS271 | Lab of Maya Schuldiner | |
| pFa6aNatN2 | Lab stock | |
| pFa6aKanMX6 | Lab stock | |
| pYM-N15 | (Janke et al, 2004) | |
| pRS426-3HA | Lab stock | |

| Reagent/Resource | Reference or Source | Identifier or Catalog Number |
|---|---|---|
| pRS426-Mpf1-3HA | This study | |
| pRs426-Tom22-HA | This study | |
| pYX142-RFP-MTS | This study | |
| pRS426-Mpf1(PH*)-3HA | This study | |
| pRS426-Flag-Pex19 | This study | |
| pYX142 | Lab stock | |
| pYX142-Tom71-HA | This study | |
| **Antibodies** | | |
| Polyclonal rat anti-HA | Roche | 11867423001 |
| Polyclonal rabbit anti-Erv2 | Lab of Roland Lill | |
| Polyclonal rabbit anti-Hexokinase | Bio-Trend | 100-4159 |
| Polyclonal rabbit anti-Tom20 | Lab stocks | |
| Polyclonal rabbit anti-Tom40 | Lab stocks | |
| Polyclonal rabbit anti-Tom70 | Lab stocks | |
| Polyclonal rabbit anti-Mcr1 | Lab stocks | |
| Polyclonal rabbit anti-Om14 | Lab stocks | |
| Polyclonal rabbit anti-Fis1 | Lab stocks | |
| Polyclonal rabbit anti-Pex14 | Lab of Ralf Erdmann | |
| Polyclonal rabbit anti-Hep1 | Lab stocks | |
| Polyclonal rabbit anti-Tom71 | Lab stocks | |
| Goat anti-rabbit IgG HRP conjugate | Bio-Rad | #1721019 |
| Goat anti-rat IgG HRP conjugate | Abcam | ab6845 |
| **Oligonucleotides and other sequence-based reagents** | | |
| OM45 C' tag pYM F | TGATAAGG GTGATGGTA AATTCTGGAG CTCGAAA AAGGACcgta cgctgcaggtcgac | This study |
| OM45 C' tag pYM R | TTATGCGGGAACCA ACCCTTTACAA TTAGCTATCTAACT Aatcgatgaattcgagctcg | This study |
| PEX3 C' tag pYM F | CAGCAACTTTGGC GTCTCCAGCTCGT TTTCCTTCAAGCCT cgtacgctgcaggtcgac | This study |
| PEX3 C' tag pYM R | ATATATTCTGGTG TGAGTGTCAGTAC TTATTCAGAGAT TAatcgatgaattcgagctcg | This study |
| OM45 WT CHK F | ATCTTCAATTG GGGGTTTAG | This study |
| OM45 WT CHK R | TCTCTACCA AACTCCTGTGC | This study |
| PEX3 WT CHK F | TGAGCAAGCTT TCCTATCAC | This study |
| PEX3 WT CHK R | CTTTCGATAC ATTGGGTCAG | This study |

| Reagent/Resource | Reference or Source | Identifier or Catalog Number |
|---|---|---|
| DLT_Tom71_Nat_Fwd | GTATATATCTCT ACATACTTGT ATATACCGAA CATAAGAA GCTCTTATG gcg cgc cag atc tgt tta g | This study |
| DLT_Tom71_Rev | CCAGTATTAAC TAAAAGTATATA TTTGACCAATA CCTGACATATCT TCTA gag ctc gat tac aac agg tg | This study |
| CHK Tom71 del F | ATGGCCGAAA ACTCCCTCCTGA | This study |
| CHK Tom71 del R | AAG CAT GCC TTT AGC CCT ATA ACG AGC TA | This study |
| GPD_Tom71_Nat F | TGTATATATCT CTACATACTTGTA TATACCGAACATAAG AAGCTCTTATGc gtacgctgcaggtcgac | This study |
| GPD_Tom71_Nat R | AGGATGGCCA CTTTGTTCTTGGTGATA AACCTCAGGA GGGAGTTTTCGGCc atcgatgaattctctgtcg | This study |
| CHK GPD Tom71 F | GAC CCA CGC ATG TAT CTA TC | This study |
| CHK GPD Tom71 R | CAG GTC TGC AGC GAG GAG | This study |
| GPD_Tom70_Nat F | AGATTCGGAAGT GAAATTACAG CTCACATCT AGGTTCTCAATTG CCAATGcgtacg ctgcaggtcgac | This study |
| GPD_Tom70_Nat R | GCAGCAACGGTTG CCAAAATGGCTG TCTTGTTCCT TGTAATGAAGC TCTTcatcgatgaattc tctgtcg | This study |
| CHK GPD Tom70 F | GAA AGA AACAC TGTGCAGGC | This study |
| CHK GPD Tom70 R | CAG GTC TGC AGC GAG GAG | This study |
| Mpf1- SACI F | GGGgagctcATG TCC TCC AGC ATC TTT GAA ATG AC | This study |
| Mpf1- XMAI R | GGGcccgggT CATAAATTC GTAAGGTCG TTAGTTC | This study |
| Ynl PHD K144A_K147A F | GGTTTCTATGG CAATGGAAGC ACTATCGCC | This study |
| Ynl PHD K144A_K147A R | GGCGATAGTGC TTCCATTGC CATAGAAACC | This study |

| Reagent/Resource | Reference or Source | Identifier or Catalog Number |
|---|---|---|
| Ynl PHD K157A F | ATGCTTCTTCTG CATTATG GAACA | This study |
| Ynl PHD K157A R | TGTTCCATAA TGCAGAAGAAGCAT | This study |
| Tom71 HA F AvrII | GGGcctaggAT GGCCGAAAAC TCCCTCCTGA | This study |
| Tom71 HA R SALI | GGGgtcgacCT AAGCGTAATCT GGAACATCG TATGGGTA AAGCATGCCTTT AGCCC | This study |
| Flag Pex19 F EcoRI | GGGgaattc ATG GAC TAC AAA GAC GAT GAC GAC AAGAATGA AAACGAGTA CGATAATT | This study |
| Flag Pex19 R BamHI | GGGggatccTTA TTGTTGTTTGC AACCGTC | This study |
| Flag Mpf1 F XmaI | GGGcccgggAT GTCCTCCAGCAT CTTTGAAATGAC | This study |
| Flag Mpf1 R SacI | GGGgagctcTCACT TGTCGTCATCGTC TTTGTAGT CTAAATTCGTAAG GTCGTTAGTTC | This study |
| Actin 1F qPCR | TGGTGATGA AGCTCAATCCAAG | This study |
| Actin 1R qPCR | TGGATGGAA ACGTAGAAGGC | This study |
| Ynl 1F qPCR | TCGCCCTTT GAAAATGCTTCT | This study |
| Ynl 1R qPCR | TCGTCTCGTG CTATTTCCCC | This study |
| **Chemicals, Enzymes and other reagents** | | |
| Zymolyase-20T | Nacalai Tesque | 07663-04 |
| HEPES | Carl Roth | HN78.1 |
| EDTA | Carl Roth | X986.2 |
| Histodenz | Sigma-Aldrich | D2158-100G |
| Fatty acid-free BSA | Sigma-Aldrich | A7030-100G |
| Percoll | Sigma-Aldrich | P1644-100ML |
| Cycloheximide | Sigma-Aldrich | C7698-5G |
| MOPS | Carl Roth | 6979.4 |
| RNA isolation kit | Macherey-NAGEL | REF 740933.50 |
| GoScript™ Reverse Transcription Mix | Promega | A2791 |
| **Software** | | |
| AxioVision | Zeiss | |
| Image J | Fiji | |
| Design and Analysis software 2.6.0 | Thermo Fisher Scientific | |
| AIDA Image Analysis | Elysia-raytest GmbH | |

| Reagent/Resource | Reference or Source | Identifier or Catalog Number |
|---|---|---|
| Corel Draw | Alludo HQ | |
| **Other** | | |
| Singer RoTor bench top robot | Singer Instruments (Maya Schuldiner Lab) | |
| Spinning disk microscope Zeiss Axio Examiner Z1 | Zeiss | |

4-(2-hydroxyethyl)-1-piperazineethanesulfonic acid (HEPES), 1.2 M sorbitol, pH 7.2). Next, cells were incubated for 1 h in 20 ml sorbitol buffer containing Zymolyase. Digestion of yeast cell walls was monitored by measuring the $OD_{600}$ of small sample of cells to detect their rupturing upon addition of water.

All further steps were carried out on ice. Spheroplasts were washed twice with 20 ml sorbitol buffer and centrifuged ($1500 \times g$, 6 min, 4 °C). Then, cells were homogenized using a dounce homogenizer in a solution of 15 ml lysis buffer (5 mM 2-(N-Morpholino)-ethane sulfonic acid (MES), 0.5 mM EDTA, 1 mM KCl) containing 0.6 M Sorbitol, Proteases inhibitors cocktail (PIC), 2 mM phenylmethylsulfonyl fluoride (PMSF), pH 5.5). Cell debris were removed by two centrifugation runs ($1600 \times g$, 10 min, 4 °C). The resulting supernatant (containing mitochondria and peroxisomes) was centrifuged ($13,000 \times g$, 5 min, 4 °C) and the pellet was resuspended in lysis buffer to $OD_{600} = 4$. The organelles were loaded on top of a density gradient consisting of 415 µl of 20%, 830 µl of 25%, 415 µl of 30%, and 830 µl of 40% Histodenz in gradient buffer A (5 mM MES, 1 mM EDTA, 1 mM KCl, and 0.1% (v/v) ethanol, pH 5.5). Gradients were centrifuged in a Beckman ultracentrifuge optima XE with a swinging bucket rotor, SW 60 Ti ($100,000 \times g$, 90 min, 4 °C, acceleration 7, brake off). A total of 12 fractions with 235 µl in each were collected from the top of the gradient and mixed with 8x sample buffer (0.5 M Tris pH 6.8, 16% SDS, 80% glycerol, 8 mg/mL bromophenol blue) to a final 2x concentration. Then 5% (v/v) β-mercaptoethanol was added and the samples were heated at 95 °C. Fractions were subjected to SDS-PAGE followed by Western blotting.

## Isolation of mitochondria

Yeast cells were grown in liquid media (volume of 2–6 L) to logarithmic phase. The cells were harvested ($3000 \times g$, 5 min, RT), resuspended in DTT buffer and incubated at 30 °C for 15 min. Cells were harvested ($2000 \times g$, 5 min, RT), washed once with spheroplasting buffer (1.2 M Sorbitol, 20 mM KPI, pH 7.2), harvested again and resuspended in spheroplasting buffer with Zymolyase (6 mg/g of cells) and incubated at 30 °C for 1 h.

Further steps were carried out on ice. Spheroplasts were homogenized in homogenization buffer (0.6 M Sorbitol, 10 mm Tris, pH 7.4, 1 mM EDTA, 0.2% fatty acid-free BSA with 2 mM PMSF) using a dounce homogenizer to obtain a cell lysate. Cell debris and nuclei were removed by two clarifying spins ($2000 \times g$, 10 min, 4 °C). The supernatant (cytosol + organelles) was centrifuged ($18,000 \times g$, 15 min, 4 °C) to pellet crude mitochondria. The resulting post nuclear supernatant (PNS) consisted of ER/microsomal and cytosolic fractions. The crude mitochondria were washed twice with SEM buffer (250 mM Sucrose, 1 mM EDTA, 10 mM MOPS) containing 2 mM PMSF and were pelleted again ($18,000 \times g$, 15 min, 4 °C).

## Subcellular fractionation

All the steps were carried out at 4 °C. Whole cell lysate and crude mitochondria were obtained as described above. To further purify mitochondria from potential contaminants, the mitochondrial fraction was layered on a Percoll gradient (25% Percoll, 2 M sucrose, 100 mM MOPS/KOH pH 7.2, 100 mM EDTA, 200 mM PMSF) and centrifuged ($80,000 \times g$, 45 min, 4 °C, slow acceleration, slow brake). Highly pure mitochondria were found as a brownish layer close to the bottom of the tube and were removed carefully with a Pasteur pipette. The mitochondria were washed several times with SEM buffer containing 2 mM PMSF and pelleted again ($18,000 \times g$, 15 min, 4 °C).

To isolate ER/microsomal and cytosolic fractions, 20 ml of PNS was clarified ($18,000 \times g$, 15 min, 4 °C) and centrifuged ($200,000 \times g$, 1 h, 4 °C). The supernatant contained the cytosolic fraction, and the brownish sticky pellet (consisting of ER) was resuspended in 2 ml of SEM buffer containing 2 mM PMSF and homogenized with a dounce homogenizer. The sample was centrifuged ($18,000 \times g$, 20 min, 4 °C) to obtain ER/microsomes in the supernatant.

The obtained fractions were precipitated with chloroform-methanol mixture and the pellet was resuspended in 2x sample buffer (125 mM Tris pH 6.8, 4% SDS, 20% glycerol, 10% β-ME, 2 mg/mL bromophenol blue) to obtain protein concentration of 2 mg/ml. Samples were heated at 95 °C for 10 min and further analyzed by SDS-PAGE and immunoblotting. Reagent and Tools Table indicates the antibodies used in the current study.

## Proteinase K (PK) assay

Isolated mitochondria (100 µg) were incubated on ice for 15 min in 50 µL of either SEM buffer (untreated) or SEM buffer containing 10 µg/mL proteinase K (PK). Then, PK activity was inhibited by addition of 2 mM PMSF. The samples were centrifuged ($18,000 \times g$, 15 min, 4 °C) and the pellets were resuspended in 2x sample buffer. Samples were heated at 95 °C for 10 min and further analyzed by SDS-PAGE and immunoblotting.

## Carbonate (alkaline) extraction

Isolated mitochondria (100 µg) were resuspended on ice in 100 µL solution containing 20 mM HEPES, 2 mM PMSF and 1x PIC, pH 7.5. This was followed by the addition of 100 µL of carbonate solution (200 mM $Na_2CO_3$, 5 mM PMSF, 1x PIC) at various pH values (10.5, 11, 11.5, or 12), and further incubation for 20 min at 4 °C. Next, pellet (membrane proteins) and supernatant fraction (soluble proteins) were separated by centrifugation ($75,000 \times g$, 30 min, 4 °C). The supernatant was precipitated by trichloroacetic acid (TCA). The pellet and precipitated proteins from the supernatant were resuspended in 40 µL 2x sample buffer, heated

at 95 °C for 10 min and further analyzed by SDS-PAGE and immunoblotting.

## Protein stability assay

Yeast strains were grown to mid-logarithmic phase. For each time point, cells corresponding to $OD_{600}$ of 2 were collected and resuspended in 1 ml of media. Cycloheximide (CHX) at final conc. of 0.1 mg/ml was added at time=0 and the cells were incubated further at 30 °C for different time periods. Then, cells were harvested ($3000 \times g$, 5 min, room temperature (RT)) and the proteins were extracted by alkaline lysis using 0.2 M NaOH, followed by heating with 2x sample buffer at 95 °C for 10 min. The samples were analyzed by SDS-PAGE and immunoblotting.

## (Immuno) fluorescence microscopy

Yeast cells were grown on synthetic media containing 2% glucose till mid-logarithmic phase. The cells (1 ml) were centrifuged ($3000 \times g$, 5 min, RT) and the cells pellet was resuspended in 50 µl water. A small portion (5 µl) of this solution was mixed with 1% (w/v) low melting point agarose and was spread on a glass slide. Confocal spinning disc microscope was used to capture images, and they were analyzed using ImageJ (more details are given in the next section).

For immunofluorescence microscopy, a published protocol was optimized (Pemberton, 2014). Yeast cells were grown till mid-logarithmic phase and to fix them, they were incubated at 30 °C for 10 min with 1% (v/v) of 37% formaldehyde. Cells were washed twice and centrifuged ($3000 \times g$, 5 min, RT) with Phosphate buffer (100 mM $KH_2PO_4$, 37.4 mM KOH, pH 6.5). Then, cells were resuspended in DTT buffer (100 mM Tris-HCl and 100 mM DTT) and incubated for 10 min at 30 °C. Cells were then washed twice with SPC buffer (1.2 M Sorbitol, 127 mM $KH_2PO_4$, 36 mM Citric acid) and spheroplasts were produced by incubating the cells for 45 min at 30 °C in SPC buffer + Zymolyase (6 mg/gr of cells). Spheroplasts were washed with SPC buffer and centrifuged ($2000 \times g$, 5 min, 4 °C) and the pellets were resuspended in 100 µl SPC buffer, snap frozen, and stored at −80 °C.

Glass slides with 15 wells were treated with 0.1% (w/v) Poly-L-Lysine for 15 min at RT to enhance cell attachment. The Poly-L-lysine was washed off by gently passing a stream of distilled water and the slides were air-dried. Next, 5 µl of spheroplasts solution were added to each well and were allowed to attach for 15 min. Excess liquid was removed, the slides were immersed in ice-cold MeOH for 5 min and were moved up and down 2–3 times. To further permeabilize the cell membrane, the slides were then immersed in acetone for 30 sec. Following this, the slides were air-dried and placed in a humid box for further steps. The cells were blocked with blocking buffer (PBS, 2% (w/v) milk, 0.1% (v/v) Tween-20) for 10 min at RT. The blocking solution was discarded, and cells were incubated at dark with 5 µg/ml primary antibody in the blocking buffer for 2 h at RT. Excess liquid was aspirated, and the cells were washed 3 times with PBS before mounting with 80% (v/v) glycerol. Cells were imaged using spinning disk microscope Zeiss Axio Examiner Z1 with a CSU-X1 real-time confocal system (Visitron) and SPOT Flex charge-coupled device camera (Visitron). Samples were observed using Zeiss Objective Plan-Apochromat 63×/1.4 Oil DIC M27. Images in Brightfield, GFP, and RFP

channels were acquired through AxioVision software. Subsequent cropping and merging was done using Fiji software.

## Rapid protein extraction

Cells grown to mid-logarithmic phase were harvested and resuspended such that 1 ml consisted of 2.5 units of $OD_{600}$. The corresponding cell pellets were resuspended in 200 µl NaOH (100 mM) and incubated for 5 min at RT. The cells were centrifuged ($3000 \times g$, 5 min, RT), resuspended in 2x sample buffer and heated for 5 min at 95 °C. The samples were centrifuged ($3000 \times g$, 5 min, RT) and the supernatant was analyzed using SDS-PAGE followed by immunoblotting.

## Pull-down assays

Cell pellets from 500 ml cultures were resuspended in 5 ml DTT buffer and incubated for 15 min in a 30 °C shaker. Cells were harvested ($2000 \times g$, 5 min, RT), washed once with spheroplasting buffer (1.2 M Sorbitol, 20 mM KPI, pH 7.2), resuspended in spheroplasting buffer with Zymolyase (6 mg/g of cells), and then incubated for 1 h in a 30 °C shaker. Cell pellets were resuspended in 1 ml lysis buffer (Tris-buffered saline (TBS), 2 mM PMSF, 1x EDTA free protease inhibitor cocktail, 5 mM EDTA) and homogenized using a douncer. Whole cell lysate (WCL) corresponding to 3 mg proteins was solubilized with 1% Triton X-100 and incubated in an overhead shaker for 30 min at 4 °C. The solubilized sample was centrifuged ($30,000 \times g$, 30 min, 4 °C) to clear out cell debris and the supernatant was incubated with anti-HA magnetic beads for 2 h at 4 °C. The beads were washed with wash buffer (TBS, 0.5% Triton X-100, 5 mM EDTA, 350 mM NaCl) and the bound proteins were eluted by incubating the beads for 10 min at 55 °C with 2x sample buffer supplemented with 0.05% $H_2O_2$. Eluted material was supplemented with 5% β-ME, incubated at 95 °C for 5 min and analyzed by SDS-PAGE followed by immunoblotting.

## Real-time quantitative PCR

RNA from 10 mL yeast culture was isolated using a mini kit for RNA isolation (Macherey-NAGEL, REF 740933.50). Next, 2.5 mg of RNA was used to prepare cDNA using the reagents and program mentioned in GoScript™ Reverse Transcription Mix, Oligo(dT) Protocol (Promega, A2791). RT-qPCR was set up in Thermo Fisher Scientific QuantStudio™ 5 Real-Time PCR System and the results were analyzed using Design and Analysis software 2.6.0. Primers used for the qPCR are listed in Reagent and Tools Table and Actin was used as a reference.

# Data availability

This study includes no data deposited in external repositories. The source data of this article can be found in the BioStudies database, accession number: S-BSST1871 (https://www.ebi.ac.uk/biostudies/studies/S-BSST1871).

The source data of this paper are collected in the following database record: biostudies:S-SCDT-10_1038-S44319-025-00440-6.

## Peer review information

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

## Acknowledgements

We thank E. Kracker for excellent technical assistance, H. Meyer and A. Fadel for help with the high-throughput screen, Marie Helmke and Luca Brenner for help with cloning, W. Girzalsky and R. Erdmann for anti-Pex14 antibody, and KS Dimmer for helpful discussions. This work was supported by the Deutsche Forschungsgemeinschaft (RA 1028/11-1 to DR and MS), the German-Israel Foundation (grant I-1458-412.13/2018 to MS and DR), the Elisabeth and Franz Knoop-Foundation (fellowship to DGV), and the Israel Science Foundation (grant 914/22 to MS and EZ). All targeting work in the Schuldiner lab is also supported by an ERC CoG from the European Union (OnTarget, 864068). The robotic system of the Schuldiner lab was purchased through the kind support of the Blythe Brenden-Mann Foundation. MS is an Incumbent of Dr. Gilbert Omenn and Martha Darling Professorial Chair in Molecular Genetics.

## Author contributions

**Nitya Aravindan**: Data curation; Investigation; Visualization; Methodology; Writing—original draft. **Daniela G Vitali**: Conceptualization; Data curation; Investigation; Visualization; Methodology. **Julia Breuer**: Validation; Investigation; Methodology. **Jessica Oberst**: Investigation; Methodology. **Einat Zalckvar**: Conceptualization; Data curation; Supervision; Visualization; Methodology. **Maya Schuldiner**: Conceptualization; Funding acquisition; Validation; Visualization; Project administration; Writing—review and editing. **Doron Rapaport**: Conceptualization; Supervision; Funding acquisition; Investigation; Writing—original draft; Project administration; Writing—review and editing.

Source data underlying figure panels in this paper may have individual authorship assigned. Where available, figure panel/source data authorship is listed in the following database record: biostudies:S-SCDT-10_1038-S44319-025-00440-6.

## Funding

## Disclosure and competing interests statement

The authors declare no competing interests.

# Expanded View Figures

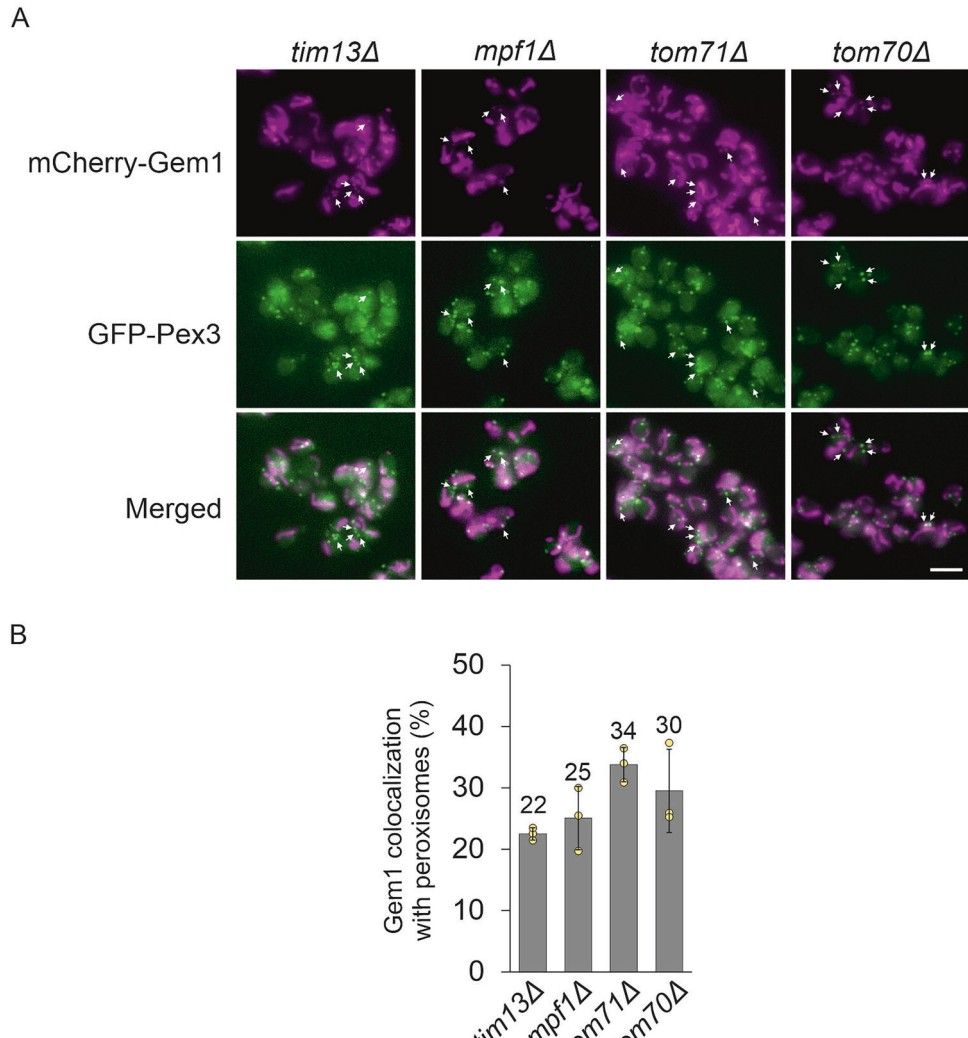

**Figure EV1. The hits influencing Fis1's distribution to mitochondria and peroxisomes impact also the dual distribution of another TA protein, Gem1.**

(A) Representative images of three strains (mpf1Δ, tom71Δ, and tom70Δ) with altered distribution of Gem1 between mitochondria and peroxisomes. The strains co-express mCherry-Gem1 and GFP-Pex3 (as peroxisomal marker). These strains maintain normal mitochondrial morphology and peroxisomes number. tim13Δ cells were used as control. Co-localization of mCherry-Gem1 (as magenta puncta, shown with white arrows) with Pex3-GFP (as green puncta, shown with white arrows) is indicated. Scale bar, 5 µm. (B) Quantification of the co-localization of Gem1 with peroxisomes. The total number of peroxisomes (visualized by Pex3-GFP) were counted in 100 cells in three independent experiments. Subsequently, the percentage of co-localization of mCherry-Gem1 puncta with the peroxisomes was determined. Error bars represent ±SD.

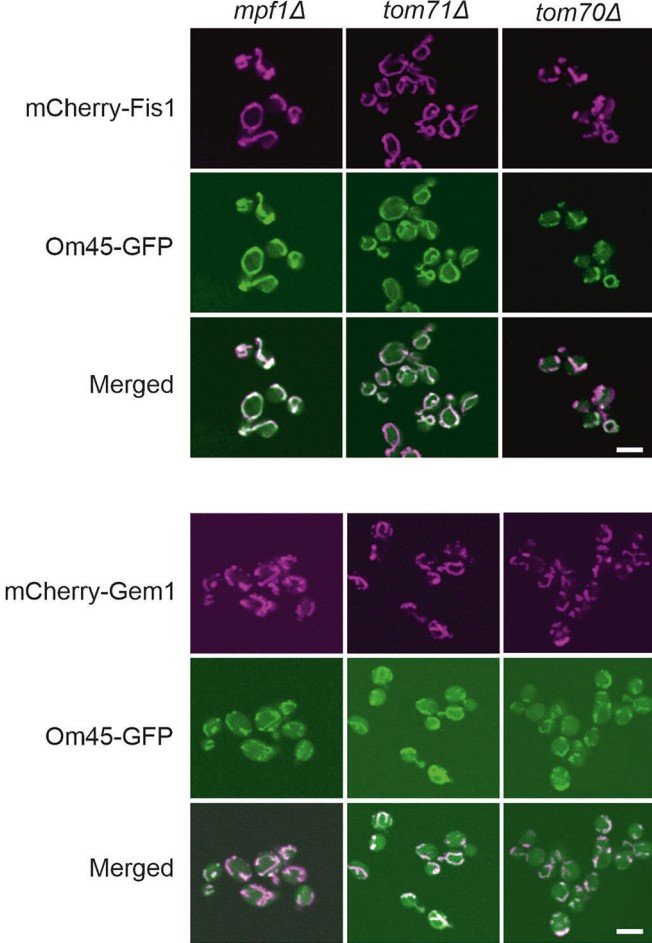

**Figure EV2. Regular mitochondrial morphology is observed in *mpf1Δ*, *tom71Δ*, and *tom70Δ* cells.**

Mitochondrial morphology was visualized by imaging of mCherry-Fis1 (top panel) and mCherry-Gem1 (bottom panel) in the indicated strains. Om45-GFP served as a marker for mitochondrial structures. Scale bar, 5 µm.

A

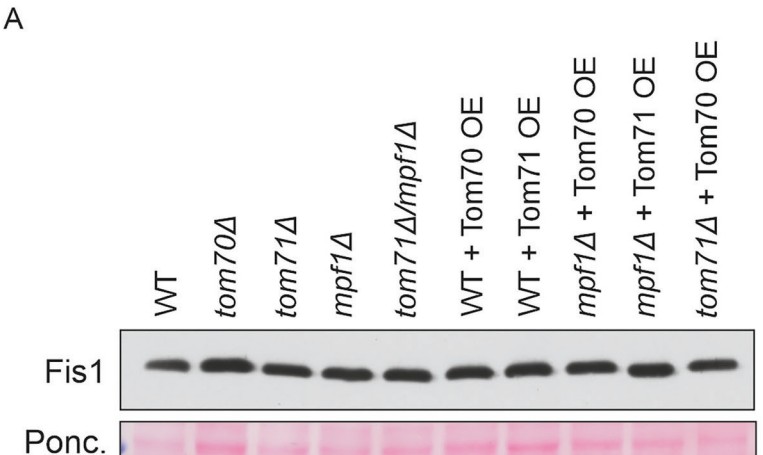

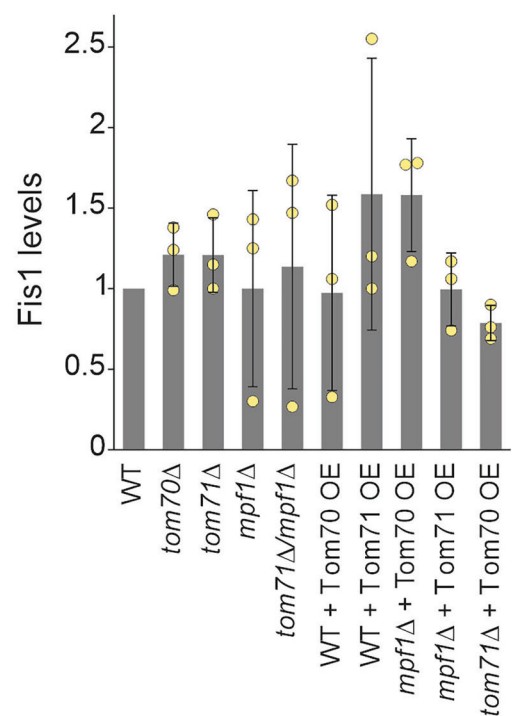

B

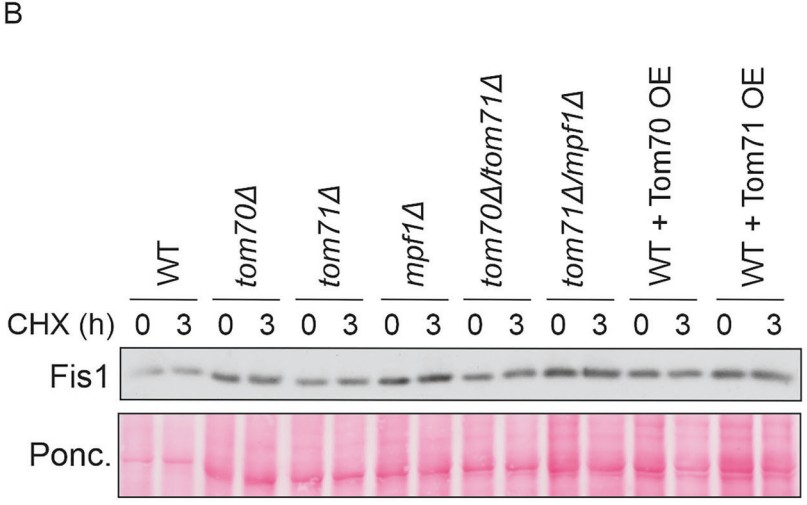

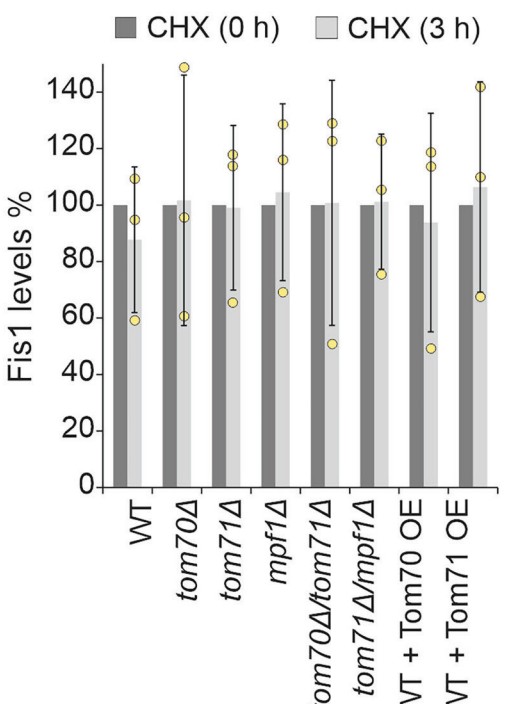

◀ **Figure EV3. The absence or overexpression of Tom70, Tom71, and Mpf1 affect neither the steady-state levels of Fis1 nor its stability.**

(**A**) Left panel: Mitochondria isolated from the indicated cells (grown on oleate-containing medium) were analyzed by SDS-PAGE and immunodecoration with anti-Fis1 antibody. The Ponceau stain is shown as loading control. Right panel: the band corresponding to Fis1 in three independent experiments as the one shown in the left panel were quantified and corrected for loading variations according to the Ponceau stain. The intensity of the band in wild type cells was set as 1. Error bars represent ±SD. (**B**) Left panel: The indicated cells were grown on glucose and then at time = 0 the translation inhibitor cycloheximide (CHX) was added and cells were further incubated for three hours. Then, cellular proteins were extracted by alkaline lysis and analyzed by SDS-PAGE and immunodecoration with an antibody against Fis1. The Ponceau stain is shown as loading control. Right panel: the bands corresponding to Fis1 in three independent experiments as the one shown in the left panel were quantified and corrected for loading variations according to the Ponceau stain. For each strain, the intensity of the Fis1 band at time = 0 was set as 100%. Error bars represent ±SD.

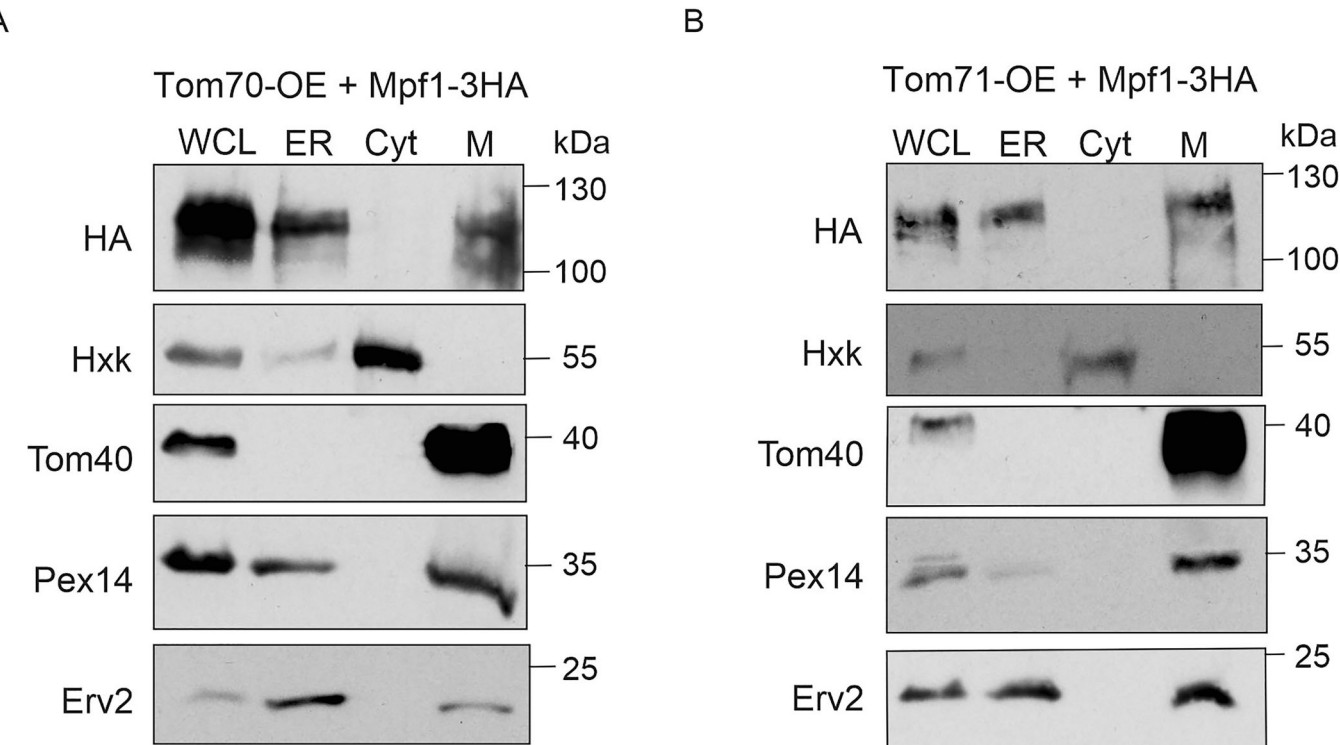

**Figure EV4. Overexpression of either Tom70 or Tom71 does not affect the subcellular location of Mpf1.**

Cells co-overexpressing Mpf1-3HA with either Tom70 (**A**) or Tom71 (**B**) were subjected to subcellular fractionation. The isolated fractions of whole cell lysate (WCL), microsomes (ER), cytosol (Cyt), and mitochondria (M) were analyzed by SDS-PAGE and immunodecoration with the indicated antibodies. Hexokinase (cytosol), Tom40 (mitochondria), Pex14 (peroxisomes), and Erv2 (ER) were used as marker proteins.

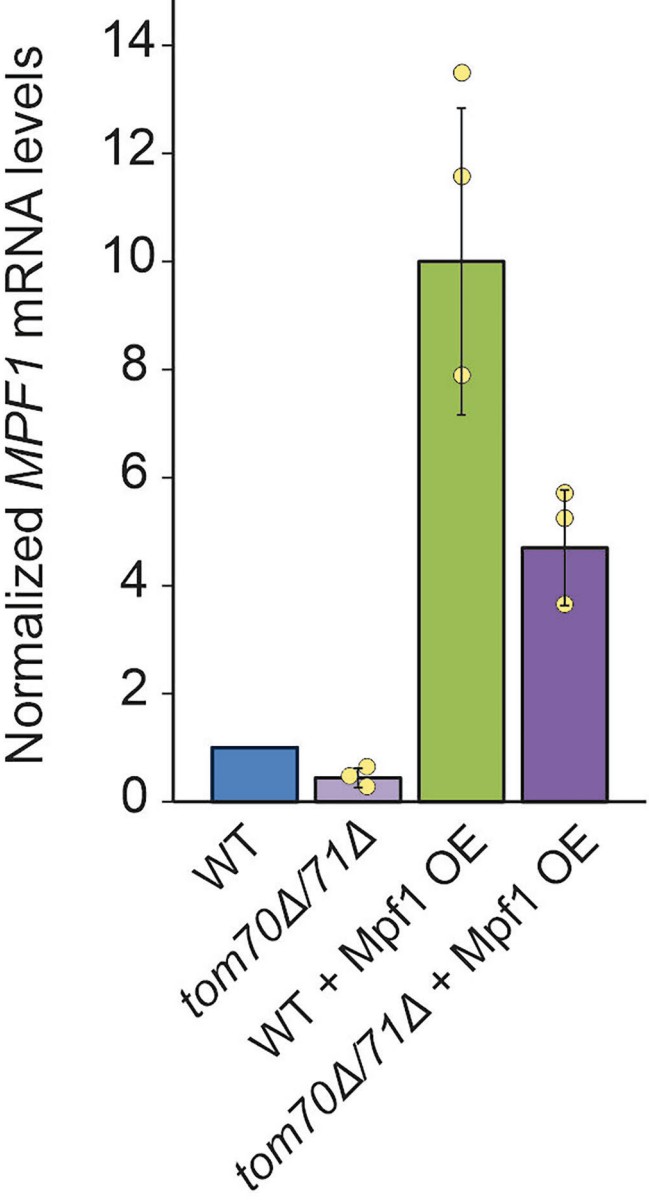

**Figure EV5.  The transcript levels of Mpf1 are reduced in *tom70/71Δ* cells.**

RT-qPCR analysis was performed in both WT and *tom70/71Δ* cells to detect transcript levels of either endogenous MPF1 or upon transformation of cells with overexpression plasmid encoding MPF1-3HA. The transcript levels of ACT1 (encoding the abundant protein Actin) served as a reference. The results of three independent experiments are depicted. Error bars represent ±SD.

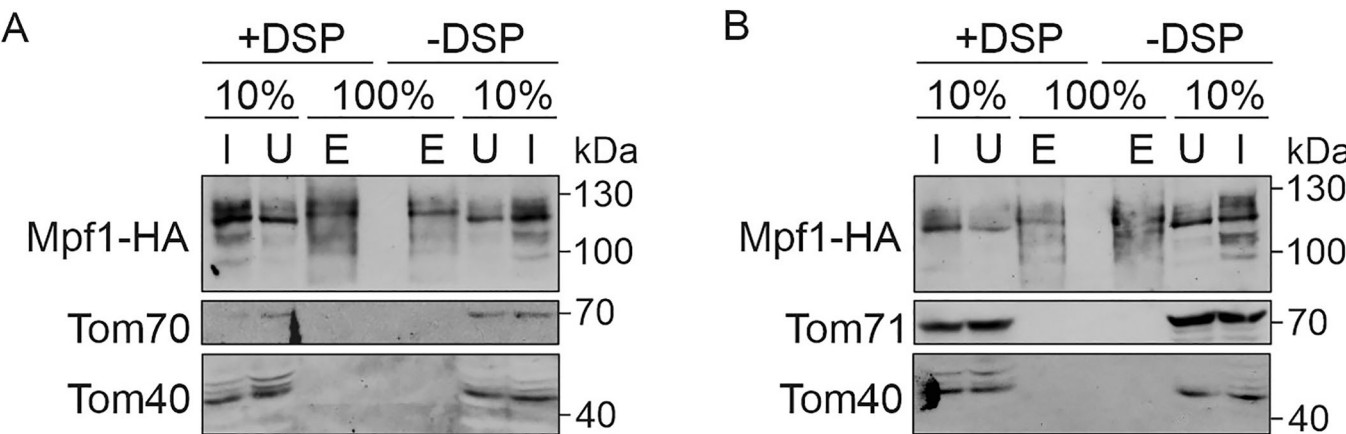

**Figure EV6. Absence of detected interaction of Mpf1 to either Tom70 or Tom71.**

(A, B) WT cells co-expressing Mpf1-HA and either Tom70-Flag (A) or Tom71-Flag (B) were lysed with 1% Triton X-100 and incubated with HA-beads in the absence (−DSP) or presence of the chemical crosslinker DSP (+DSP). Fractions representing the input (I, 10%), unbound material (U, 10%), and the eluate (E, 100%) were analyzed by SDS-PAGE and immunodecoration with the indicated antibodies.

