## [Peer Review File · EMBO Reports]

Mpf1 affects the dual distribution of tail-anchored proteins between mitochondria and peroxisomes

Nitya Aravindan, Daniela Vitali, Julia Breuer, Jessica Oberst, Einat Zalckvar, Maya Schuldiner, and Doron Rapaport

Corresponding author(s): Doron Rapaport (doron.rapaport@uni-tuebingen.de)

Review Timeline:

Transfer Date:	8th Nov 24
Editorial Decision:	17th Jan 25
Revision Received:	17th Feb 25
Editorial Decision:	24th Feb 25
Revision Received:	26th Feb 25
Accepted:	11th Mar 25

Editor: *Martina Rembold*

Transaction Report: This manuscript was transferred to EMBO reports following peer review at Review Commons.

Review #1

1. Evidence, reproducibility and clarity:

Evidence, reproducibility and clarity (Required)

Summary: This paper attempts to address how dual targeting of certain tail-anchored proteins, Fis1 and Gem1, occurs to two compartments, namely peroxisomes and mitochondria. Not much is known regarding the regulation of this dual distribution, so the authors undertook a genome-wide yeast screen for mutants that would favor preferential sorting of Fis1 and Gem1 to peroxisomes, relative to mitochondria, using wild-type yeast as control. The absence of one of three proteins, Mpf1, Tom70 and Tom71, resulted in a higher proportion of TA proteins at peroxisomes rather than at mitochondria. Mpf1 was shown to be very unstable and localized peripherally on the mitochondrial outer membrane facing the cytosol. Tom71, in comparison with its more highly expressed paralog, Tom70, had a unique role in favoring Fis1 distribution at mitochondria. The presence of Tom70/71 was required to maintain normal Mpf1 levels. Tom70 and 71 seem to act differently.

Overexpression of Tom70 rescues and normalizes the Fis1 distribution defect of mpf1 cells (excessive proportional Fis1 sorting to peroxisomes), suggesting that the role of Mpf1 can be fulfilled by Tom70 overexpression. However, this rescue was only partial in the absence of Tom71, suggesting a unique role for Tom71 in Fis1 distribution. Tom71 overexpression, on the other hand, led to more Fis1 in mitochondria, in both WT and mpf1 cells.

The conclusion is that factors that skew the dual targeting of certain TA proteins and insights into the regulation of dual targeting of TA proteins have been identified.

Major issues

1. It is unclear if these proteins are involved in sorting, stabilization of organelle pools, or quality control of mis-sorted Fis1 and Gem1. Experiments designed to clarify this will be necessary and helpful.
2. If Fis1 was targeted exclusively to one or other compartment, that would affect the normal biogenesis and/or morphology of peroxisomes and mitochondria. This makes me wonder whether the screening criteria could have missed more interesting mutants that skewed the distribution more profoundly, like pex19 with no peroxisomes. This should be explained.
3. The screen and the data in Table S1 seem qualitative, rather than quantitative. Please explain if this is a misunderstanding or provide a rationale.
4. The skewing of normal targeting could be viewed as an aspect of mistargeting, which is a

known phenomenon for TA proteins like Pex15, or quality control because the steady-state levels of yeast mitochondrial TA proteins, such as the small TOM complex subunits, Tom5, Tom6, and Tom7, are reduced upon deletion TOM receptors (PMID 31945731). As another example, the depletion of Pex19 caused more Fis1 and Gem1 on mitochondria, and the ATPase, Msp1, is involved in the quality control of mistargeted peroxisomal TA proteins, yet these proteins did not show up in the screen. Are the proteins here involved directly in targeting or abrogation of mistargeting or quality control?

5. What is the direct link between the targeted proteins and Mpf1, Tom 70 and Tom 71? Why does overexpression of either Tom70 or Tom71 rescue fully or partially the skewing of Fis1 distribution to peroxisomes seen in i cells. Without addressing these questions, it is hard to decide if there is a direct or indirect role of these targeting factors.

6. What is the physiological significance of the rapid turnover of Mpf1, if any?

7. I disagree with the claim that "our findings contribute to novel insights on factors responsible for regulating the dual distribution of Fis1 to mitochondria and peroxisomes" in that I fail to see the novelty without knowledge of the mechanism.

****Easily addressable points****

1. Fig. 2 shows, which addresses the skewing of the targeting of TA proteins requires a loading control.

2. Although the authors claim that mpf1 cells grow better than WT in oleate, the data in Fig 4 is not quantitative. Why is the growth of mpf1 mildly higher only in Soleate and YPO? The higher growth of mpf1 in oleate is speculated to be due to more peroxisomes but this point needs experimental support in YPO and Soleate.

2. Significance:

Significance (Required)

The study is preliminary, without insightful understanding why or how dual distribution of TA proteins is regulated. In this sense, this is a preliminary report, worthy indeed of further study, but falling short of providing a deeper mechanistic understanding of the problem or generality of the principles of dual targeting of proteins to different subcellular compartments.

is is aimed at a broad cell biology audience

3. How much time do you estimate the authors will need to complete the suggested revisions:

Estimated time to Complete Revisions (Required)

(Decision Recommendation)

More than 6 months

4. Review Commons values the work of reviewers and encourages them to get credit for their work. Select 'Yes' below to register your reviewing activity at Web of Science Reviewer Recognition Service (formerly Publons); note that the content of your review will not be visible on Web of Science.

Yes

Review #2

1. Evidence, reproducibility and clarity:

Evidence, reproducibility and clarity (Required)

Aravindan and colleagues address a very timely question how the dual localization of proteins to different cell organelles is controlled. They used an elegant genetic screen to study which factors affect the localization of Fis1 between peroxisomes and mitochondria. They identified Tom70, Tom71 and the previously uncharacterized Mpf1. They found that deletion or overexpression of these proteins affect the distribution of Fis1 between mitochondria and peroxisomes. The authors demonstrated that Mpf1 binds to the mitochondrial surface and is a highly unstable protein. The parallel loss of Tom70 and Tom71 leads to drastic reduction of Mpf1 levels, indicating a close connection between the TOM receptors and Mpf1. Overall, the presented study provides new insights into the control of the cellular localization of the tail-anchored protein Fis1. The findings are exciting and interesting for broad readership. The quality of the experimental work is high and the results are clearly described. I have only minor concerns.

1. Do Tom70 and/or Tom71 recruit Mpf1 to mitochondria? The authors may test whether overexpression of Tom70/Tom71 affect the localization of Mpf1.
2. In their genetic screen, the authors have found several hundred hits. Subsequently, the authors made manual inspection to focus on the candidate proteins. Can the authors extend the description of the criteria they used for the manual selection?

3. Previously, Pex19, Ssa1 and Sti1 were identified as targeting factor for Fis1 (Cichocki et al., 2018). Were these proteins also identified in the genetic screen presented here? The authors should discuss a model how these factors cooperate with Tom70, Tom71 and Mpf1.

2. Significance:

Significance (Required)

Overall, the presented study provides new insights into the control of the cellular localization of the tail-anchored protein Fis1. The findings are exciting and interesting for broad readership.

3. How much time do you estimate the authors will need to complete the suggested revisions:

Estimated time to Complete Revisions (Required)

(Decision Recommendation)

Less than 1 month

No

Review #3

1. Evidence, reproducibility and clarity:

Evidence, reproducibility and clarity (Required)

Proteins that are destined to different cellular organelles are synthesized on the cytosolic ribosomes and subsequently guided to their final localizations. Most of the proteins contain organelle-specific targeting signals. However, there are some proteins that are targeted to more than one organelle, in an event called dual or multiple targeting. These

proteins frequently do not bear any specific information in their sequence. Aravindan et al. aimed to elucidate a mechanism of dual targeting of the two tail-anchored proteins in yeast, Fis1 and Gem1, that localize to either mitochondria or peroxisomes and their targeting is not based on yet-identified sequence information. The authors performed a high throughput systematic microscopy screen that allowed them to identify the protein candidates, Tom70, Tom71 and Mpf1, that affected the distribution of Fis1 between both organelles. They further validated these hits using biochemical methods. The authors found that individual deletions of Tom71 and Mpf1 led to a reduction of Fis1 in the mitochondria and a subsequent increase of Fis1 in the peroxisomes. Moreover, overexpression of Tom71 specifically increased the pool of mitochondrial and reduced the pool of peroxisomal Fis1. Overall, the authors identified the factors affecting the dual distribution of tail-anchored proteins between mitochondria and peroxisomes in yeast, a molecular basis of which was unknown.

****Major comments:****

Generally, the claims and the conclusions are supported by the data. The limitation is that the behavior of Mpf1 is analyzed upon tagging.

Also, it should be clarified whether Tom70 and/or Tom71 interact with Mpf1 for targeting of the tail-anchored models.

The transcriptomics in the absence of Tom70/Tom71 should give an interesting picture whether Mpf1 is unique in terms of its high mRNA levels, or rather belong to a group of proteins (which in turn can be of interest for understanding of protein targeting)- this suggestion is for the future. But it could be interesting to analyze the published literature for the Mpf1 expression regulation.

****Minor comments:****

Introduction, 3rd paragraph: Notably, mammalian homologues of these fission components were also found to be dually localized to mitochondria and peroxisomes lacks a citation.

2. Significance:

Significance (Required)

Identification of a new player MPf1 that acts together with Tom71 in controlling distribution of the model tail-anchored proteins.

The study identified new factors involved in the dual targeting of tail-anchored proteins between mitochondria and peroxisomes in yeast. It is the first study that reports the relevance of Tom70, Tom71 and Mpf1 in this process. Moreover, so far, no function was assigned to the Mpf1 protein. Hence, it is the first report that studies the role of this protein in the cellular context. The study does not explain the exact mechanisms by which Tom70, Tom71 and Mpf1 regulate the distribution of tail-anchored proteins between the two organelles. Thus, it remain an interesting observation ready to follow in the future.

While Tom70 was extensively studied and functionally described, the role of Tom71 was not described in such details. This study thus expands our knowledge on the functionality of Tom71. Assigning the role for Mtf1 is new.

This study is of interest to protein translocation and mitochondrial biogenesis field and can be described as basic research.

My field of expertise:

Protein translocation, mitochondria

3. How much time do you estimate the authors will need to complete the suggested revisions:

Estimated time to Complete Revisions (Required)

(Decision Recommendation)

Less than 1 month

4. Review Commons values the work of reviewers and encourages them to get credit for their work. Select 'Yes' below to register your reviewing activity at Web of Science Reviewer Recognition Service (formerly Publons); note that the content of your review will not be visible on Web of Science.

Yes

Full Revision

Manuscript number: RC-2024-02500

Corresponding author(s): Doron, Rapaport

1. General Statements

Dear Editor,

We would kindly ask you to consider the enclosed manuscript entitled “Mpf1 is a novel factor that affects the dual distribution of tail-anchored proteins between mitochondria and peroxisomes” by Aravnidan et al. for publication in EMBO Reports.

In this contribution, we address an understudied and timely question: How is dual localization of proteins to different cell organelles regulated? For that aim, we employed a genetic and visual high throughput screen with yeast collections to study which factors affect the localization of the tail-anchored proteins Fis1 and Gem1 between peroxisomes and mitochondria. We identified Tom70, Tom71 and the previously uncharacterized protein Mpf1. We characterized this novel protein and analyzed the contribution of the three proteins to the dual-distribution process. Collectively, our findings contribute to novel insights on factors responsible for regulating the dual distribution of proteins to mitochondria and peroxisomes

This contribution was submitted originally to Review Commons. Considering its topic and the positive feedback that we got from the reviewers, we believe that this work is best suited to be published in EMBO Reports. In a recent EMBO Workshop, I discussed a potential submission to EMBO Reports with Dr. Martina Rembold (Senior Scientific Editor) who encouraged me to do so. Thus, please find enclosed the revised version of our manuscript where we included additional experiments and further controls as well as optimized and expanded the text.

As outlined in the “Response to Reviewers” section, we have addressed all points raised by the three reviewers. Thus, we hope that you will find our revised version suitable for publication in EMBO Reports.

We thank you for your consideration.

Yours sincerely,

Doron Rapaport

We are very thankful to all three reviewers for the productive and thoughtful comments and suggestions that helped us to improve our contribution.

Reviewer #1

Major issues

1. It is unclear if these proteins are involved in sorting, stabilization of organelle pools, or quality control of mis-sorted Fis1 and Gem1. Experiments designed to clarify this will be necessary and helpful.

To address the questions raised by the reviewer, we performed additional experiments. First, we monitored the steady-state levels of Fis1 in strains where the levels of Tom70, Tom71, or Mpf1 are altered. The new results, which are depicted in revised Figure S3A demonstrate that neither deletion of these proteins nor their overexpression cause a significant change in the steady-state levels of Fis1. Second, we tested the stability of Fis1 in the abovementioned strains. We did not observe any noteworthy alterations in the stability of Fis1 upon manipulating the levels of Tom70, Tom71, or Mpf1 (revised Figure S3B). Thus, our accumulative data suggests that these latter proteins are not involved in the stability and/or quality control of Fis1 but rather contribute to its dual targeting.

2. If Fis1 was targeted exclusively to one or other compartment, that would affect the normal biogenesis and/or morphology of peroxisomes and mitochondria. This makes me wonder whether the screening criteria could have missed more interesting mutants that skewed the distribution more profoundly, like pex19 with no peroxisomes. This should be explained.

We fully agree with the reviewer that in the current study our criteria for selection of hits for further investigations might cause missing of some interesting hits. Indeed, we intend in future studies to inspect the images of our screen and to search for additional hits under alternative criteria. However, in the current study, we wanted to better understand dual distribution between mitochondria and peroxisomes and such insights can be better obtained when both organelles are still present. For example, if peroxisomes are completely absent since one of the Peroxins is deleted, it will be difficult to study dual distribution.

3. The screen and the data in Table S1 seem qualitative, rather than quantitative. Please explain if this is a misunderstanding or provide a rationale.

The data which are included in Table S1 are qualitative and are based on manual visual inspection of the relevant images. We inspected thousands of images and could not find a reliable way to analyze the images in an automatic manner. However, in those cases where our criteria, which we explained in the text, were fulfilled, we counted the number of peroxisomes in

which the Fis1 stain co-localizes with the peroxisome stain and calculated the percentage of such peroxisomes from the total number of inspected peroxisomes. Next, we compared this number to the value found in control cells. Thus, the screen provided also quantitative information for the selected hits. An example of such analysis is given in Figure 1C.

4. The skewing of normal targeting could be viewed as an aspect of mistargeting, which is a known phenomenon for TA proteins like Pex15, or quality control because the steady-state levels of yeast mitochondrial TA proteins, such as the small TOM complex subunits, Tom5, Tom6, and Tom7, are reduced upon deletion TOM receptors (PMID 31945731). As another example, the depletion of Pex19 caused more Fis1 and Gem1 on mitochondria, and the ATPase, Msp1, is involved in the quality control of mistargeted peroxisomal TA proteins, yet these proteins did not show up in the screen. Are the proteins here involved directly in targeting or abrogation of mistargeting or quality control?

The reviewer raises valid questions that we partially address already in point #1 above. Pex19 did not appear as a hit in our screen because in the absence of Pex19 there are no peroxisomes at all. Msp1 was not detected as a hit probably because it does not recognize Fis1 or Gem1 as substrates since both proteins are normal bona fide proteins of both peroxisomes and mitochondria where Msp1 is active. The new experiments that are described in our respond to point #1 above suggest that Tom70, Tom71, or Mpf1 are not involved in quality control and/or stability of Fis1.

5. What is the direct link between the targeted proteins and Mpf1, Tom 70 and Tom 71? Why does overexpression of either Tom70 or Tom71 rescue fully or partially the skewing of Fis1 distribution to peroxisomes seen in i cells. Without addressing these questions, it is hard to decide if there is a direct or indirect role of these targeting factors.

To better address this point, we conducted experiments to investigate whether Tom70, Tom71 or Msp1 can bind directly to Fis1. To that aim, we expressed as recombinant proteins GST-Tom70, GST-Tom71, GST-Mpf1, and GST alone as control. Next, we tested the ability of these purified proteins to bind newly synthesized radiolabeled molecules of Fis1 that are translated in a cell-free system (reticulocyte lysate). Unfortunately, although we employed many different conditions, we could not detect any reproducible interactions. Obviously, these negative results might suggest that either there are no direct interactions between the tested proteins and Fis1 or these interactions are too transient to be detected.

6. What is the physiological significance of the rapid turnover of Mpf1, if any?

To better understand this issue, we performed new experiments in which we grew cells expressing Mpf1-HA on glucose-containing medium and then shifted the cells to oleate-containing medium. We observed a drastic reduction in the levels of Mpf1 after two hrs on oleate (revised Figure 5C). This observation agrees with our assumption that less Mpf1 results in more Fis1 molecules on peroxisomes, which in turn causes more peroxisomes. The elevated

number of peroxisomes can then support better growth on oleate. Further supporting the importance of Fis1 for growth on oleate are our new findings that cells lacking Fis1 show retarded growth on oleate at 37°C (revised Figure 4B). Moreover, we demonstrate in the revised version that cells grown on oleate have significantly more peroxisomes per cell as those grown on glucose and the absence of Mpf1 further increases this number (revised Figure 4C).

7. I disagree with the claim that "our findings contribute to novel insights on factors responsible for regulating the dual distribution of Fis1 to mitochondria and peroxisomes" in that I fail to see the novelty without knowledge of the mechanism.

We hope that with the new data described above we can convince the reviewer that our contribution indeed provides new insights. However, if the reviewer still disagrees with our wording, we will be happy to modify it.

Easily addressable points

1. Fig. 2 shows, which addresses the skewing of the targeting of TA proteins requires a loading control.

Our revised Figure S3A compares the steady state levels of Fis1 in the various strains used in Fig. 2 and thus, provides an equivalent for loading controls for the gradient separations. These new results show that the steady-state levels of Fis1 are basically similar among the employed strains.

2. Although the authors claim that mpf1 cells grow better than WT in oleate, the data in Fig 4 is not quantitative. Why is the growth of mpf1 mildly higher only in Soleate and YPO? The higher growth of mpf1 in oleate is speculated to be due to more peroxisomes but this point needs experimental support in YPO and Soleate.

In the revised version we included new experiments to address this point. When yeast cells are grown on oleate as the sole carbon source, they are completely dependent on peroxisomal β -oxidation for energy production. Thus, the phenotype of deleting MPF1 is most obvious on oleate. We propose that upon the deletion of MPF1 more Fis1 molecules are found at peroxisomes. This elevated number causes in turn more fission of peroxisomes and thus more peroxisomes per cell. As discussed above, more peroxisomes mean improved usage of oleate and hence better growth. In the revised version we provide new results showing that: (i) cells lacking Fis1 have growth defects specially when grown on oleate at elevated temperature (revised Figure 4B), and (ii) cells contain an elevated number of peroxisomes when they are grown on oleate. Importantly, the absence of Mpf1 further increases this number (revised Figure 4C).

Significance:

The study is preliminary, without insightful understanding why or how dual distribution of TA proteins is regulated. In this sense, this is a preliminary report, worthy indeed of further study, but falling short of providing a deeper mechanistic understanding of the problem or generality of the principles of dual targeting of proteins to different subcellular compartments.

We provide in the revised version a large body of new results and hope that these new findings will convince the reviewer of the novelty and importance of our study.

Reviewer #2

Overall, the presented study provides new insights into the control of the cellular localization of the tail-anchored protein Fis1. The findings are exciting and interesting for broad readership. The quality of the experimental work is high and the results are clearly described. I have only minor concerns.

We thank the reviewer for his/her high opinion on our study.

1. Do Tom70 and/or Tom71 recruit Mpf1 to mitochondria? The authors may test whether overexpression of Tom70/Tom71 affect the localization of Mpf1.

We followed the proposal of the reviewer and performed subcellular fractionation of cells overexpressing either Tom70 or Tom71. The new results, which are included in revised Figure S4, demonstrate that such overexpression does not affect the subcellular distribution of Mpf1. These findings suggest that neither Tom70 nor Tom71 is the limiting factor in the association of Mpf1 with mitochondria.

2. In their genetic screen, the authors have found several hundred hits. Subsequently, the authors made manual inspection to focus on the candidate proteins. Can the authors extend the description of the criteria they used for the manual selection?

It seems that there is confusion here. All the strains (ca. 5000) were inspected manually by looking at four images for each strain. Next, we picked those strains that display an unusual co-staining of Cherry-Fis1 with either mitochondria or peroxisomes. Among those initially selected strains only those that fulfilled the following two criteria were considered further as real hits: (i) strains with normal number of peroxisomes per cell, and (ii) strains with regular mitochondrial morphology. Among those strains that fulfilled both criteria, we concentrated on those that fulfilled these criteria also for mCherry-Gem1. Finally, we decided to focus on three hits: TOM70, TOM71, and MPF1. This explanation is also included in the revised Results section of our contribution

3. Previously, Pex19, Ssa1 and Sti1 were identified as targeting factor for Fis1 (Cichocki et al., 2018). Were these proteins also identified in the genetic screen presented here? The authors should discuss a model how these factors cooperate with Tom70, Tom71 and Mpf1.

We indeed found both Ssa1 and Sti1 as hits in our visual screen although surprisingly, with an opposite effect (See Table S1). Whereas the deletion of STI1 resulted in reduced signal of Fis1 on peroxisomes, the absence of Ssa1 caused an increased signal at peroxisomes. We did not follow these hits in the current contribution but intend to do so in future studies.

We did not follow the strain without Pex19 because this strain completely lacks peroxisomes. Since in the current study we wanted to better understand dual distribution between mitochondria and peroxisomes, we feel that such insights can be obtained only when both organelles are still present.

Tom70 and Tom71 are known to function as docking sites on the surface of mitochondria for cytosolic chaperones loaded with mitochondrial precursor proteins. Hence, it might be that Tom70/71 are binding to the chaperones Ssa1/Sti1 while the latter are loaded with newly synthesized tail-anchored proteins. These ideas are included in our revised Discussion section.

Significance:

Overall, the presented study provides new insights into the control of the cellular localization of the tail-anchored protein Fis1. The findings are exciting and interesting for broad readership.

We are very thankful to the reviewer for his/her support of our work.

Reviewer #3

Overall, the authors identified the factors affecting the dual distribution of tail-anchored proteins between mitochondria and peroxisomes in yeast, a molecular basis of which was unknown.

We thank the reviewer for recognizing the novelty of our work.

Major comments

Generally, the claims and the conclusions are supported by the data. The limitation is that the behavior of Mpf1 is analyzed upon tagging.

We are aware of the limitation by using a tagged variant of Mpf1. However, since we do not have a working antibody against this protein, there is no other option for its detection by Western blotting. The used HA-tag is rather a small one (only 9 amino acid residues) and according to our experience, it rarely affects the targeting/stability/function of a protein.

Also, it should be clarified whether Tom70 and/or Tom71 interact with Mpf1 for targeting of the tail-anchored models.

We followed this suggestion and performed pull-down assays with a tagged version of Mpf1 in the presence or absence of a chemical cross-linker (DSP). Although we tried several different experimental set-ups, we could not detect any considerable interaction between Mpf1 and either Tom70 or Tom71. Since these are negative results that do not exclude detection of the potentially transient interactions under other experimental conditions, we decided not to include these findings in the revised version.

Full Revision

The transcriptomics in the absence of Tom70/Tom71 should give an interesting picture whether Mpf1 is unique in terms of its high mRNA levels, or rather belong to a group of proteins (which in turn can be of interest for understanding of protein targeting)- this suggestion is for the future. But it could be interesting to analyze the published literature for the Mpf1 expression regulation.

We thank the reviewer for this interesting suggestion that we will follow in future studies. Meanwhile, we looked at the SPELL database to identify genes with similar expression profiles like MPF1. The GO terms that were enriched among those genes are “energy reserve metabolic process” and “polysaccharide metabolic process”. We aim to look at the future at such possible connections.

Minor comments

Introduction, 3rd paragraph: Notably, mammalian homologues of these fission components were also found to be dually localized to mitochondria and peroxisomes lacks a citation.

The text of this sentence was modified, and a reference was added.

Significance:

Identification of a new player MPf1 that acts together with Tom71 in controlling distribution of the model tail-anchored proteins.

The study identified new factors involved in the dual targeting of tail-anchored proteins between mitochondria and peroxisomes in yeast. It is the first study that reports the relevance of Tom70, Tom71 and Mpf1 in this process. Moreover, so far, no function was assigned to the Mpf1 protein. Hence, it is the first report that studies the role of this protein in the cellular context. The study does not explain the exact mechanisms by which Tom70, Tom71 and Mpf1 regulate the distribution of tail-anchored proteins between the two organelles. Thus, it remains an interesting observation ready to follow in the future.

While Tom70 was extensively studied and functionally described, the role of Tom71 was not described in such details. This study thus expands our knowledge on the functionality of Tom71. Assigning the role for Mtf1 is new.

This study is of interest to protein translocation and mitochondrial biogenesis field and can be described as basic research.

We thank the reviewer for his/her positive evaluation of our contribution. We already included in the revised version more experiments that shed light on mechanistical details and aim to add even more insights in future studies.

Dear Doron,

Thank you for the submission of your manuscript to EMBO Reports. Please accept my apologies regarding the delay in handling it. It was seen again by two of the original referees and we have only now received the second report (copied below my signature). As you will see, both referees are positive about its publication pending textual changes and the inclusion of negative data in the manuscript.

I will list below the guidelines for formatting and information on additional files and information we will need.

Once your revised manuscript has been submitted we will perform a number of quality checks on it. I list below a few items that you can already take care of in order to speed up the quality checks.

Among others, a team of data editors will check the figure legends for completeness. Going through them myself I noticed a number of points that will need your attention:

- In graphs: please show the individual datapoints in addition to the mean and error bars. E.g., Fig. 1C should display the average of each independent experiment in addition to the average across all three. In Fig. 4C we would need the values for all cells counted in Fig. 4C in the form of a scatter blot.

- Figure 4C lacks a scale bar.

- 'n' needs to be specified for Fig. 5D and 8B

- Fig. 7B and 7C lack information on the variation between the three colonies that were measured.

Please check all other figure legends along these lines.

- Figure S1 - S5 could be supplied as Expanded View figures (see point 6 below). The nomenclature is Figure EV# and the legend needs to be provided in the manuscript file following the main figure legends.

- Materials and Methods are called "Methods" and we need a Reagent and Tools table (see point 12).

- Regarding the Author Contributions, we now use CRediT to specify the contributions of each author in the journal submission system. CRediT replaces the author contribution section, which therefore needs to be removed from the manuscript text. You can use the free text box in our system if you wish to provide more detailed descriptions. See also guide to authors <https://www.embopress.org/page/journal/14693178/authorguide#authorshipguidelines>.

- References: please use et al after the 10th author and remove the DOIs from the reference list.

- Please upload the Supplemental Table as a dataset (Dataset EV1). Please change the name also in the .xls file itself and provide the legend in a separate tab.

- Table S2 - S5 should be part of the Reagents and Tools table. If you prefer to keep the longer Table S3 separate, please call it Table EV1.

- Finally, EMBO Reports papers are accompanied online by

A) a short (1-2 sentences) summary of the findings and their significance,

B) 2-3 bullet points highlighting key results and

C) a schematic summary figure that provides a sketch of the major findings (not a data image).

Please provide the summary figure as a separate file in PNG or JPG format at a size of 550x300-600 pixels (width x height).

Please note that the size is rather small and that text needs to be readable at the final size. Please send us this information along with the revised manuscript.

GENERAL INFORMATION ON THE REVISION:

2) individual production quality figure files as .eps, .tif, .jpg (one file per figure).

Please download our Figure Preparation Guidelines (figure preparation pdf) from our Author Guidelines pages <https://www.embopress.org/page/journal/14693178/authorguide> for more info on how to prepare your figures.

4) a complete author checklist, which you can download from our author guidelines (). Please insert information in the checklist that is also reflected in the manuscript. The completed author checklist will also be part of the RPF.

5) Please note that all corresponding authors are required to supply an ORCID ID for their name upon submission of a revised manuscript (). Please find instructions on how to link your ORCID ID to your account in our manuscript tracking system in our Author guidelines
()

6) We replaced Supplementary Information with Expanded View (EV) Figures and Tables that are collapsible/expandable online. A maximum of 5 EV Figures can be typeset. EV Figures should be cited as 'Figure EV1, Figure EV2' etc... in the text and their respective legends should be included in the main text after the legends of regular figures.

- Please include a dedicated "Data Availability" section at the end of the Methods (suggested wording: "The [structural coordinates | microarray | mass spectrometry] data from this publication have been deposited to the [name of the database] database [URL] and assigned the identifier [accession | permalink | hashtag]."). Should this not apply, this should still be stated as "This study includes no data deposited in external repositories."

Additional information on source data and instruction on how to label the files are available .

10) Figure legends and data quantification:

- the name of the statistical test used to generate error bars and P values,
- the number (n) of independent experiments (please specify technical or biological replicates) underlying each data point,
- the nature of the bars and error bars (s.d., s.e.m.)

- If the data are obtained from n {less than or equal to} 5, show the individual data points in addition to the SD or SEM.

- If the data are obtained from n {less than or equal to} 2, use scatter blots showing the individual data points.

11) Our journal encourages inclusion of *data citations in the reference list* to directly cite datasets that were re-used and obtained from public databases. Data citations in the article text are distinct from normal bibliographical citations and should directly link to the database records from which the data can be accessed. In the main text, data citations are formatted as follows: "Data ref: Smith et al, 2001" or "Data ref: NCBI Sequence Read Archive PRJNA342805, 2017". In the Reference list, data citations must be labeled with "[DATASET]". A data reference must provide the database name, accession number/identifiers and a resolvable link to the landing page from which the data can be accessed at the end of the reference.

Further instructions are available at .

12) All Materials and Methods need to be described in the main text using our 'Structured Methods' format. According to this format, the Methods section includes a Reagents and Tools Table (listing key reagents, experimental models, software and relevant equipment and including their sources and relevant identifiers) followed by a Methods and Protocols section describing the methods, ideally using a step-by-step protocol format. The aim is to facilitate adoption of the methodologies across labs. Please download and fill our Reagents and Tools Table template (.docx), which you can find in our author guidelines:

13) As part of the EMBO publication's Transparent Editorial Process, EMBO Reports publishes online a Review Process File to accompany accepted manuscripts. This File will be published in conjunction with your paper and will include the referee reports, your point-by-point response and all pertinent correspondence relating to the manuscript.

Kind regards,

Martina

=====

Referee #1:

The authors carefully addressed my concerns in the revised version of the manuscript. The presented study identified with Mpf1, Tom70 and Tom71 factors that affect the distribution of proteins between mitochondria and peroxisomes. The conclusions are well based on a set of elegant experimental approaches. The work represents an important contribution to our understanding how cellular distribution of proteins is regulated and is therefore interesting for a broad readership. I recommend publication of this manuscript in EMBO Rep..

Referee #2:

I am fully satisfied with the way how the authors addressed previous critique and reviewed the article accordingly.

I have only two suggestions for the authors:

- I recommend to substantiate the current statement on the lack of direct interaction between Mpf1 and Tom70/Tom71. This can be done either by addition of experimental work to the supplement, or by expanding what exactly was done (similarly to the respective content in the rebuttal). I personally support including the experiments which fall into story logics but are negative.

- while reading the revised version of the MS, which is very exciting, I suggest to highlight the role of Tom71 slightly stronger. For example, more excessive discussion of the roles assigned experimentally to Tom70 vs Tom71 would be nice for a less specialized reader.

Referee #1:

The authors carefully addressed my concerns in the revised version of the manuscript. The presented study identified with Mpf1, Tom70 and Tom71 factors that affect the distribution of proteins between mitochondria and peroxisomes. The conclusions are well based on a set of elegant experimental approaches. The work represents an important contribution to our understanding how cellular distribution of proteins is regulated and is therefore interesting for a broad readership. I recommend publication of this manuscript in EMBO Rep..

Referee #2:

I am fully satisfied with the way how the authors addressed previous critique and reviewed the article accordingly.

I have only two suggestions for the authors:

- I recommend to substantiate the current statement on the lack of direct interaction between Mpf1 and Tom70/Tom71. This can be done either by addition of experimental work to the supplement, or by expanding what exactly was done (similarly to the respective content in the rebuttal). I personally support including the experiments which fall into story logics but are negative.

- while reading the revised version of the MS, which is very exciting, I suggest to highlight the role of Tom71 slightly stronger. For example, more excessive discussion of the roles assigned experimentally to Tom70 vs Tom71 would be nice for a less specialized reader.

Rev_Com_number: RC-2024-02500

New_manu_number: EMBOR-2024-60743V1-T

Corr_author: Rapaport

Title: Mpf1 is a novel factor that affects the dual distribution of tail-anchored proteins between mitochondria and peroxisomes

Addressing the Reviewers' comments

Referee #1:

The authors carefully addressed my concerns in the revised version of the manuscript. The presented study identified with Mpf1, Tom70 and Tom71 factors that affect the distribution of proteins between mitochondria and peroxisomes. The conclusions are well based on a set of elegant experimental approaches. The work represents an important contribution to our understanding how cellular distribution of proteins is regulated and is therefore interesting for a broad readership. I recommend publication of this manuscript in EMBO Rep..

We are grateful to the reviewer for his/her positive opinion on our contribution.

Referee #2:

I am fully satisfied with the way how the authors addressed previous critique and reviewed the article accordingly.

We are grateful to the reviewer for his/her positive opinion on our contribution.

I have only two suggestions for the authors:

- I recommend to substantiate the current statement on the lack of direct interaction between Mpf1 and Tom70/Tom71. This can be done either by addition of experimental work to the supplement, or by expanding what exactly was done (similarly to the respective content in the rebuttal). I personally support including the experiments which fall into story logics but are negative.

According to the reviewer's proposal, we added Appendix Fig. S1 to the modified version that presents the relevant pull-down experiment. This figure demonstrates that under both conditions namely, the presence or absence of the chemical cross-linker DSP, we could not detect any interaction between Mpf1-HA and either Tom70 or Tom71.

- while reading the revised version of the MS, which is very exciting, I suggest to highlight the role of Tom71 slightly stronger. For example, more excessive discussion of the roles assigned experimentally to Tom70 vs Tom71 would be nice for a less specialized reader.

As the reviewer suggested, we added to the Discussion section of the modified version new text that deals with the known functions of Tom71 and Tom70.

Manuscript number: EMBOR-2024-60743V2

Title: Mpf1 is a novel factor that affects the dual distribution of tail-anchored proteins between mitochondria and peroxisomes

Author(s): Nitya Aravindan, Daniela Vitali, Julia Breuer, Jessica Oberst, Einat Zalckvar, Maya Schuldiner, and Doron Rapaport

Dear Doron,

Thank you for the submission of your revised manuscript to EMBO Reports. We have now completed all editorial checks on it and noted a few minor corrections that are needed. I am now writing with an 'accept in principle' decision, which means that I will be happy to accept your manuscript for publication once these minor issues/corrections have been addressed, as follows.

- Please reduce the number of keywords to 5.
- Please place the Disclosure and competing interests statement after the Acknowledgments.
- Appendix: since it is only one figure, you could also change it to Figure EV6. If you prefer to keep the Appendix, we need a title page with a table of contents and page numbers (even if it is only one page...).
- Author Checklist: please complete the information on Corresponding author name, journal and manuscript ID# (top left, line 4 - 6).
- Please enter the following funding information in the online manuscript tracking system: the Elisabeth and Franz Knoop-Foundation; ERC CoG from the European Union (OnTarget, 864068); the Blythe Brenden-Mann Foundation.
- Please provide a callout to Dataset EV1. (I also noted a typo in the legend/header: systeamtic).
- There is a callout to Figure S2 on page 5, but such a figure is absent. Please correct the callout.
- As per our editorial policies, all data supporting conclusions must be part of the manuscript. In this respect, I note that the reported effect on oleate upon deleting TOM71 or TOM70 is based on 'data not shown'. Please either include the relevant data or remove the statement.
- Figure legends:
 - *) Please note that information related to n is missing in the legend of figure EV5
 - *) Although 'n' is provided, please describe the nature of entity for 'n' in the legends of figures 7B, D; 8B.
 - *) Please note that the error bars are not defined in the legends of figures 8B, EV1 B, EV3 A, B, EV5
- As a standard procedure we modify title and abstract to make them more accessible to a general audience. Please find my suggestions below my signature.

Once you have made these minor revisions, please use the following link to submit your corrected manuscript:

Link Not Available

If all remaining corrections have been attended to, you will then receive an official decision letter from the journal accepting your manuscript for publication in the next available issue of EMBO reports. This letter will also include details of the further steps you need to take for the prompt inclusion of your manuscript in our next available issue.

Thank you for your contribution to EMBO Reports.

Kind regards,

Martina

=====

Title and abstract:

Mpf1 affects the dual distribution of tail-anchored proteins between mitochondria and peroxisomes

Most cellular proteins require targeting to a distinct cellular compartment to function properly. A subset of proteins is distributed to two or more destinations in the cell and little is known about the mechanisms controlling the process of dual/multiple targeting. Here, we provide insight into the mechanism of dual targeting of proteins between mitochondria and peroxisomes. We perform a high throughput microscopy screen in which we visualize the location of the model tail-anchored proteins Fis1 and Gem1 in the background of mutants in virtually all yeast genes. This screen identifies three proteins, whose absence results in a higher portion of the tail-anchored proteins in peroxisomes: the two paralogues Tom70, Tom71, and the uncharacterized gene YNL144C that we rename mitochondria and peroxisomes factor 1 (Mpf1). We characterize Mpf1 to be an unstable protein that associates with the cytosolic face of the mitochondrial outer membrane. Furthermore, our study uncovers a unique contribution of Tom71 to the regulation of dual targeting. Collectively, our study reveals, for the first time, factors that influence the dual targeting of proteins between mitochondria and peroxisomes.

e r a a n f r a n s s e s e r e r e s e e a r s

Prof. Doron Rapaport
University of Tuebingen
Interfaculty Institute of Biochemistry
Auf der Morgenstelle 34
Tuebingen 72076
Germany

Dear Doron,

I am very pleased to accept your manuscript for publication in the next available issue of EMBO reports. Thank you for your contribution to our journal.

Kind regards,

Martina
